# A model for microbial interactions and metabolomic alterations in *Candida glabrata-Staphylococcus epidermidis* dual-species biofilms

**Maria Michela Salvatore**[1,2], **Angela Maione**[3], **Alessandra La Pietra**[3], **Federica Carraturo**[3], **Alessia Staropoli**[2,4], **Francesco Vinale**[2,5,6], **Anna Andolfi**[1,6], **Francesco Salvatore**[3], **Marco Guida**[3,6]*, **Emilia Galdiero**[3]*

**1** Department of Chemical Sciences, University of Naples Federico II, Naples, Italy, **2** Institute for Sustainable Plant Protection, National Research Council, Portici, Italy, **3** Department of Biology, University of Naples Federico II, Naples, Italy, **4** Department of Agricultural Sciences, University of Naples Federico II, Portici, Italy, **5** Department of Veterinary Medicine and Animal Productions, University of Naples Federico II, Naples, Italy, **6** BAT Center—Interuniversity Center for Studies on Bioinspired Agro-Environmental Technology, University of Naples Federico II, Portici, Italy

* marco.guida@unina.it (MG); emilia.galdiero@unina.it (EG)

**Data Availability Statement:** All relevant data are within the paper.

## Abstract

The fungus *Candida glabrata* and the bacterium *Staphylococcus epidermidis* are important biofilm-forming microorganisms responsible of nosocomial infections in patients. In addition to causing single-species disease, these microorganisms are also involved in polymicrobial infections leading to an increased antimicrobial resistance. To expand knowledge about polymicrobial biofilms, in this study we investigate the formation of single- and dual-species biofilms of these two opportunistic pathogens employing several complementary approaches. First, biofilm biomass, biofilm metabolic activity and the microbial composition in single- and dual-species biofilms were assessed and compared. Then, the expression of three genes of *C. glabrata* and three genes of *S. epidermidis* positively related to the process of biofilm formation was evaluated. Although *S. epidermidis* is a stronger biofilm producer than *C. glabrata*, both biological and genetic data indicate that *S. epidermidis* growth is inhibited by *C. glabrata* which dominates the dual-species biofilms. To better understand the mechanisms of the interactions between the two microorganisms, a broad GC-MS metabolomic dataset of extracellular metabolites for planktonic, single- and dual-species biofilm cultures of *C. glabrata* and *S. epidermidis* was collected. As demonstrated by Partial Least Squares Discriminant Analysis (PLS-DA) of GC-MS metabolomic data, planktonic cultures, single- and dual-species biofilms can be sharply differentiated from each other by the nature and levels of an assortment of primary and secondary metabolites secreted in the culture medium. However, according to our data, 2-phenylethanol (secreted by *C. glabrata*) and the synergistically combined antifungal activity of 3-phenyllactic acid and of the cyclic dipeptide *cyclo*-(L-Pro-L-Trp) (secreted by *S. epidermidis*) play a major role in the race of the two microorganisms for predominance and survival.

**Funding:** The authors received no specific funding for this work.

**Competing interests:** The authors have declared that no competing interests exist.

## Introduction

Biofilms are communities of microorganisms that attach to abiotic or biotic surfaces, encased in an extracellular matrix and often characterized by a polymicrobial nature.

Compared to planktonic cells, microbial cells in mature biofilms display different phenotypes exhibiting accumulation of extracellular matrix material and increased resistance to drugs. In fact, biofilms have a highly recalcitrant behaviour to resist to antimicrobial drugs and host defences which make them difficult to treat [1, 2]. Because of this, infections associated with biofilm formation on tissues and implanted prosthetic devices can often only be resolved by removing the infected tissue or the prosthesis by means of invasive surgery. An understanding of factors which control mechanisms of microbial biofilm development may provide strategies for prevention and/or non-surgical treatment of biofilms associated infections.

Due to the complexity of biological and metabolic interactions, the association of both eukaryotic and prokaryotic organisms represents a particularly challenging problem in the area of polymicrobial biofilm research [3, 4] which has been approached in many different ways involving advanced analytical and biological techniques.

A recent very promising approach, which is eminently suitable to explore the complexities involved in the interactions between a fungus and a bacterium in dual-species biofilms, consists in the integration of biological evidence with comparative transcriptomics and untargeted metabolomics data [5].

Due to its frequent involvement in polymicrobial biofilm-based infections [1, 2], *Candida albicans* represents the most common species of the genus *Candida* studied in coexistence with bacterial pathogens and the only one for which the metabolic mechanisms behind the mixed biofilm formations were investigated. In fact, an altered carbohydrates, amino acids, polyamine and lipid metabolisms have been observed for dual-species biofilms of *C. albicans*/*Proteus mirabilis* [6] and *C. albicans*/*Klebsiella pneumoniae* [7] compared to single species biofilm.

Even if *C. albicans* represents the most prevalent fungal biofilm-forming pathogen, other non-*albicans Candida* species are a serious health threat worldwide [8]. Among them, *Candida glabrata*, is responsible for 10–20% of invasive candidiasis [9–11]. It is a commensal of the mucous membranes of healthy individuals and may be an opportunistic pathogen causing severe diseases in the immunodeficient host. *Candida glabrata* forms a biofilm, embedded in the extracellular matrix, on both infected tissues and inert surfaces, which is considered one of the main virulence factors for infection and persistence in the host. In addition, this fungus possesses acquired resistance to azoles [12].

*Candida glabrata* share the same habitat in humans with *Staphylococcus epidermidis* which is the most abundant microorganism on human skin with biofilm producing property and responsible for urinary tract infections [13]. Hence, the co-occurrence of *C. glabrata* and *S. epidermidis* could promote mixed nosocomial infections of these pathogens prolonging hospitalization and increasing mortality.

As part of our interests, aimed at exploring interspecies interactions during polymicrobial biofilm development, the biological and metabolic alterations associated with the formation of *C. glabrata*/*S. epidermidis* biofilms were assessed for the first time by evaluating data acquired on planktonic, single-species and dual-species biofilms cultures of the targeted microorganisms. Safranin-O staining/de-staining assay and XTT reduction assay were employed for biofilm biomass quantification, and colony forming units (CFUs) were enumerated in order to quantify the abundance of each species within dual biofilms [14, 15]. Further information on interactions of *C. glabrata* and *S. epidermidis* during biofilm development, was gathered by evaluating, by quantitative Real-Time PCR, the expression of some important genes associated

with adhesion, biofilm formation and virulence of *C. glabrata* (i.e., *Erg11*, *ALS3*, FKS*1*) and *S. epidermidis* (i.e., *icaD*, *FnbA*, *EbpS*). Finally, we combined untargeted metabolomic footprint analysis of data acquired by Gas Chromatography-Mass Spectrometry (GC-MS), on single and dual-species biofilms and planktonic cultures, to delineate how interspecies interactions modulate the metabolome of each organism. On this basis we developed a model which provides key insights into biological and metabolic interactions in *C. glabrata/S. epidermidis* biofilms and which paves the way for future studies targeted to specific strains, surfaces and culture media mimicking medical devices and biochemical environment of infected tissues.

## Materials and methods

### Strains and culture conditions

*Candida glabrata* DSM11226 strain and a clinical isolate from human skin of *Staphylococcus epidermidis*, molecularly identified at 99% and compared with a reference strain *S. epidermidis* strain NRLFFD301, were used in this study and maintained in YPD (1% *w/v* yeast extract, 2% *w/v* peptone, 2% *w/v* glucose, 1.5% Agar) and tryptone soya agar plates (TSA) (OXOID, Basingstoke, UK) respectively. For subsequent experiments with mono- and dual-species, the strains were sub-cultured in tryptone soya broth (TSB) (OXOID, Basingstoke, UK) supplemented with glucose 1% *w/v* to reach finally $10^6$ cells $\cdot$ mL$^{-1}$ in phosphate buffered saline (PBS) (OXOID, Basingstoke, UK).

### In vitro biofilm analysis

**Biofilm formation.** Biofilms were grown in 96-well polystyrene plates as previously described [16]. Briefly, cultures were washed twice in PBS and 100 μl cultures diluted to $10^6$ cells $\cdot$ mL$^{-1}$ in TSB supplemented with 1% glucose for single-species biofilms and 100 μl with final inoculum $10^6$ cells $\cdot$ mL$^{-1}$ (mixed 1:1) for dual-species were inserted into 96-well plates and incubated at 37˚C for 24 or 48 h.

**Biofilm quantification.** All biofilms were rinsed once with PBS to remove non-adhered cells, and safranin-O staining/de-staining and XTT reduction assays were employed, respectively, to quantify the total biofilm biomass and metabolic activity of biofilm cells, as previously described [14, 15].

For biofilm biomass estimate with safranin-O, biofilms were carefully washed twice with PBS and stained with a 0.1% safranin-O (3,7-diamino-2,8-dimethyl-5-phenylphenazin-5-ium chloride, Sigma Aldrich, MO, USA) aqueous solution for 15 min at 25˚C. Wells were carefully washed three times with PBS to remove excess stain and air dried at room temperature. Aqueous acetic acid solution (10% *v/v*) was added to the wells containing stained biofilms and incubated for 20 min at 25˚C to de-stain and the absorbance measured at 530 nm.

Cell metabolic activity was determined using a 2, 3-bis (2-methoxy-4-nitro-5-sulfo-phenyl)-2H-tetrazolium-5 carboxanilide (XTT) based kit (Kit cell Counting Kit-8 Enzolife Science, Switzerland). Wells were washed and incubated for 3 h with 10 μl XTT reagent at 37˚C. The resulting absorbance was read at 450 nm.

Absorbance was read using a microplate reader (SYNERGY H4 BioTek, BioTek Instruments, Agilent Technologies, Winooski, VT 05404, USA).

Data were reported as mean of three independent experiments ± standard deviation. Analysis of variance (ANOVA) followed by Tukey's multiple comparison test was applied and values of *p*-value < 0.05 were considered statistically different.

**Viability of *C. glabrata* and *S. epidermidis* in single- and dual- biofilm.** The viabilities of *C. glabrata* and *S. epidermidis* in single and mixed biofilms, were measured counting colony forming units per well (CFU·well$^{-1}$) both at 24 and 48 h. Briefly, the formed biofilms were

scraped off from each well and put into a microtube filled with 200 μL TSB medium + 1% glucose to obtain a biofilm suspension. For *S. epidermidis*, the biofilm suspension was gradually diluted and inoculated on TSA plates; for co-cultures, inocula were supplemented with 0.05 mg·mL$^{-1}$ amphotericin B (Sigma-Aldrich, St. Louis, MO, United States). For *C. glabrata*, the gradually diluted biofilm suspension was inoculated on YPD agar plates; for co-cultures, inocula were supplemented with 1 mg·mL$^{-1}$ ampicillin (Sigma-Aldrich, St. Louis, MO, United States). Agar plates were incubated at 37˚C aerobically and colonies were counted the day after. Experiments were performed in triplicate and results are presented as mean ± 95% confidence limits.

**Quantitative Real-Time PCR (qRT-PCR).** Three sets of transcriptomic analyses were performed with *S. epidermidis* and *C. glabrata* planktonic cells and single/dual- biofilms. Total RNA was isolated using a Direct-zolTM RNA Miniprep Plus Kit (ZYMO RESEARCH, Irvine, CA, USA) according to the manufacturer's instructions, and the purity and concentration of the extracted RNA were verified using Nanodrop spectrophotometer 2000 (Thermo Scientific Inc., Waltham, MA, USA). 1000 ng of total RNA was retrotranscribed with an iScriptTM cDNA Synthesis kit (Bio-Rad, Milan, Italy), following the manufacturer's instructions. qRT-PCR was used to determine the expressions of various biofilm-related genes (*Erg11*, *ALS3*, *FKS1* for *C. glabrata* and *icaD*, *FnbA*, *EbpS* for *S. epidermidis*) and the specific primers used are listed in Table 1. Two housekeeping genes, *act* and *16s* rRNA were used, respectively. The qRT-PCR was done in an AriaMx Real-Time PCR instrument (Agilent Technologies, Inc., Santa Clara, CA, USA), SYBR Green Supermix, ROX (Quantabio, Beverly, MA) was used, and fluorescence was measured using Agilent Aria 1.7 software (Agilent Technologies, Inc.) according to the manufacturer's instructions. At least two independent cultures were used. The final reaction volume was 10 μl, containing 1 μl of cDNA, 1 μl of the forward and reverse primers, 5 μl of master mix and 3 μl of deionized water and the temperature cycle included initial denaturation for 3 minutes at 95˚C followed by 40 cycles of 15 s at 95˚C and 45 s at 60˚C. The final stage was selected to be at 95˚C for 15 s, 60˚C for 1 minute and 95˚C for 15 s to draw melting curves. Gene quantitative relative expression was examined using the REST

**Table 1. List of primer sequences employed to determine the expression of biofilm-related genes via qRT-PCR.**

| Protein | Gene | Primer | Sequence (5'→3') |
|---|---|---|---|
| Actin | *actin* | *C. glabrata_ACT1_F* | TTGCCACACGCTATTTTGAG |
| | | *C. glabrata_ACT1_R* | ACCATCTGGCAATTCGTAGG |
| Ergosterol biosynthesis enzyme | ERG11 | *C. glabrata_ERG11_F* | ATTGGTGTCTTGATGGGTGGTC |
| | | *C. glabrata_ERG11_R* | TCTTCTTGGACATCTGGTCTTTCA |
| Agglutinin like-sequence 3 | ALS3 | *C. glabrata_ALS3_F* | CTGGACCACCAGGAAACACT |
| | | *C. glabrata_ALS3_R* | GGTGGAGCGGTGACAGTAGT |
| Beta-1,3-glucan synthase catalytic subunit | FKS1 | *C. glabrata_FKS1_F* | GTTGCAGTCGCTACATTGCTA |
| | | *C. glabrata_FKS1_R* | TAGCGTTCCAGACTTGGGAA |
| 16S *ribosomial* RNA | 16S | *S. epidermidis_16SrRNA_F* | TACATGCAAGTCGAGCGAAC |
| | | *S. epidermidis_16SrRNA_R* | AATCATTTGTCCCACCTTCG |
| Poly-beta-1,6-N-acetyl-D-glucosamine synthesis protein IcaD | icaD | *S. epidermidis_icaD_F* | ACCCAACGCTAAAATCATCG |
| | | *S. epidermidis_icaD_R* | GCGAAAATGCCCATAGTTTC |
| Fibronectin-binding protein A | FnbA | *S. epidermidis_FnbA_F* | AAATTGGGAGCAGCATCAGT |
| | | *S. epidermidis_FnbA_R* | GCAGCTGAATTCCCATTTTC |
| Elastin-binding protein EbpS | EbpS | *S. epidermidis_EbpS_F* | GGTGCAGCTGGTGCAATGGGTGT |
| | | *S. epidermidis_EbpS_R* | GCTGCGCCTCCAGCCAAACCT |

software (Relative Expression Software Tool, Weihenstephan, Germany, version 1.9.12) based on the Pfaffl method [17, 18]. Fold changes larger than 2 or lower than 0.5 were considered significant.

## Metabolomic analysis

**Sample preparation and GC-MS analysis.** For analysis of extracellular metabolites from planktonic and biofilm cultures of *C. glabrata* and *S. epidermidis*, 100 microliters of the liquid culture medium from each planktonic or biofilm culture were collected and centrifugated ($5000 \times g$, 10 min). After centrifugation, supernatants were carefully dried with a stream of nitrogen at room temperature, and metabolites in the residue were derivatized with *N*,*O*-bis (trimethylsilyl)-trifluoroacetamide (BSTFA) (Fluka, Buchs, Switzerland), as previously described [19, 20]. Each experiment was performed in triplicate to obtain three biological replicates for each treatment (or class) defined by the combination of cultured microorganism (*C. glabrata* or *S. epidermidis* or dual species), the type of culture (planktonic or biofilm) and the temperature (24 h or 48 h). The GC-MS analyses of the resulting trimethylsilyl derivatives of metabolites were carried out using an Agilent 8890 GC instrument (Milan, Italy) coupled to an Agilent 5977B Inert MS. For the separation a HP-5MS capillary column ((5%-phenyl)-methyl-polysiloxane stationary phase) was housed in the GC oven. The following oven GC temperature program was employed: 70°C for 1 min, 10°C·min$^{-1}$ until the column temperature reached 170°C, and 30°C·min$^{-1}$ until the column temperature reached 280°C, 280°C for 8 min. The solvent delay was set to 5 min. Analyses were carried using helium as carrier gas at a flow rate of 1 mL·min$^{-1}$ and the GC injector was set in splitless mode at 250°C. Measurements were conducted under electron impact (EI) ionization (70 eV) in full scan mode (*m/z* 35–600) at a frequency of 3.9 Hz. The EI ion source and quadrupole mass filter temperatures were kept, respectively, at 230 and 150°C. GC-MS analysis of each of the three biological replicates for a given treatment was replicated three times (giving 3×3 = 9 GC-MS data files for each treatment).

**Data processing and statistical analysis.** GC-MS data were deconvoluted using the National Institute of Standards and Technology (NIST) program Automated Mass Spectral Deconvolution and Identification System (AMDIS) [21, 22], and then "conserved" metabolites across each biological and technical replicate were listed and tracked using SpectConnect program [23]. Technically, in the context of SpectConnect software, a conserved metabolite is one that consistently persists in replicate samples (at least in 75% of the observations in one class or condition).

Multivariate and univariate analyses of GC-MS data were performed with our in-house.m scripts in MATLAB R2021a (Mathworks, Natick, MA, USA) [24].

First, the relative abundances matrix (RA matrix) created by SpectConnect program was reduced to exclude non identified signals and the reduced 54×106 RA matrix of identified metabolites employed for all successive evaluations.

For multivariate statistical analyses, i.e., Principal Component Analysis (PCA) and Partial Least Squares Discriminant Analysis (PLS-DA), the reduced RA matrix of identified metabolites was standardized so that each column had zero mean and unit variance.

Univariate analysis of data in the RA matrix was performed in order to compare metabolites levels, one by one, in a selected pair of classes. This allows one to determine metabolites whose levels in the extracellular medium are significantly different between two selected conditions (treatments). Within this framework, for each conserved metabolite, two arrays of RA values (one array for each of the two compared conditions) are extracted from the RA matrix and passed to the MATLAB *ttest2* function. This function compares the two vectors using

unpaired Student's *t*-test and returns, between other statistics, a *p*-value. After this, for each metabolite, a fold change (FC) value is calculated as the ratio of the RA averages in the two compared vectors. A metabolite is considered to be differentially expressed between two compared classes if the associated *p*-value is lower than 0.05 (i.e., the significance threshold is set to 5%) and the associated fold change is greater than 2 (upregulated) or lower than 0.5 (downregulated).

The identification of metabolites was performed by matching their deconvoluted EI mass spectra at 70 eV with those stored in the NIST 20 mass spectral library [25] and the Golm metabolome database [26, 27]. Furthermore, the identification was supported by the Kovats retention index (RI) calculated for each metabolite by the Kovats equation using the standard *n*-alkane mixture in the range C7–C40 (Sigma-Aldrich, Saint Louis, MO, USA) analysed under the same conditions [28].

## Results

### In vitro biofilm analysis

Formation of single- and dual-species microbial biofilms in conventional 96-wells polystyrene microtiter plates was assessed, after 24 h and 48 h incubation at 37˚C, by two complementary assays. First, staining/de-staining of biofilm cells with safranin-O and absorbance measurements on a microplate reader (safranin-O assay) was directed to obtain a relative estimate of total biofilm biomass production in different biofilm cultures [29]. Second, an XTT reduction assay was employed to compare cells metabolic activity in different biofilms by measuring the absorbance, on a microplate reader, of the orange-coloured product resulting from the reduction of XTT by enzymes that are only present in metabolically active live cells [30].

Within this frame, either from Fig 1A or 1B, it can be seen that *S. epidermidis* is a stronger biofilm producer than *C. glabrata*. From the point of view of safranin-O and XTT assays, both

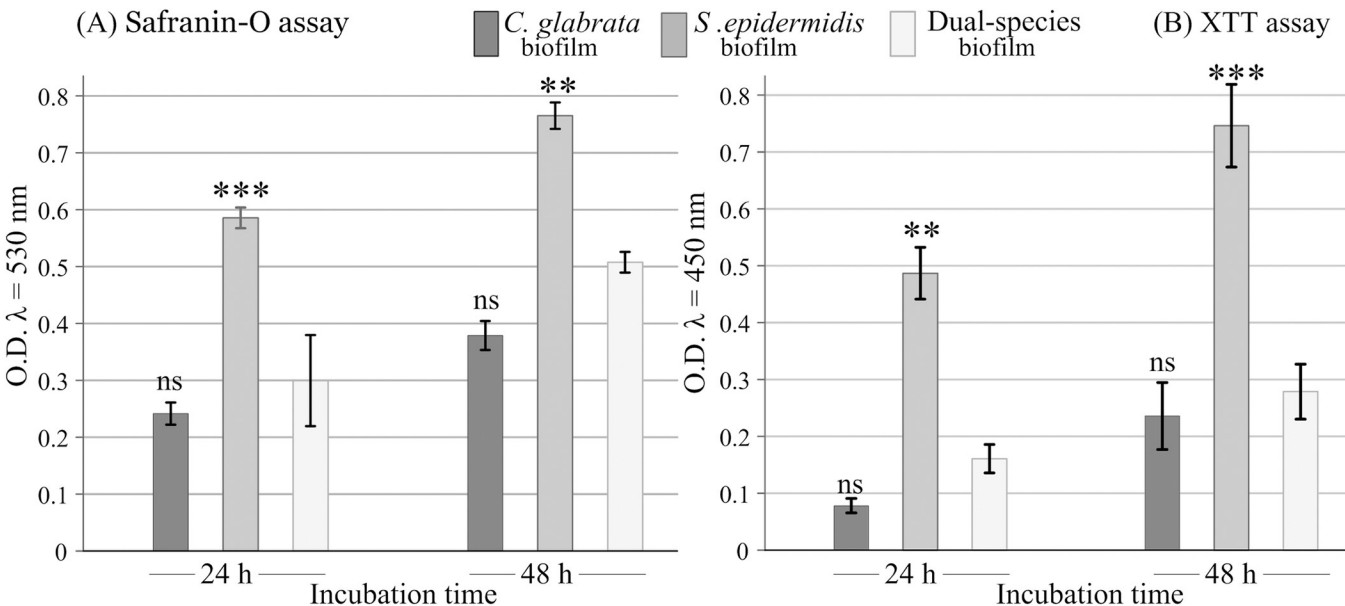

**Fig 1.** *C. glabrata* (dark grey bars) and *S. epidermidis* (medium grey bars) single- and dual-species (light grey bars) biofilm characterization at 24 h and 48 h by (A) Safranin-O assay and (B) XTT assay. All tests were performed in triplicate and results are presented as the mean ± standard deviation. At each incubation time, asterisks indicate significant differences vs. dual-species biofilm (light grey bars) with Tuckey's multi-comparison test (** = *p*-value < 0.01; *** = *p*-value < 0.001; ns = not significant).

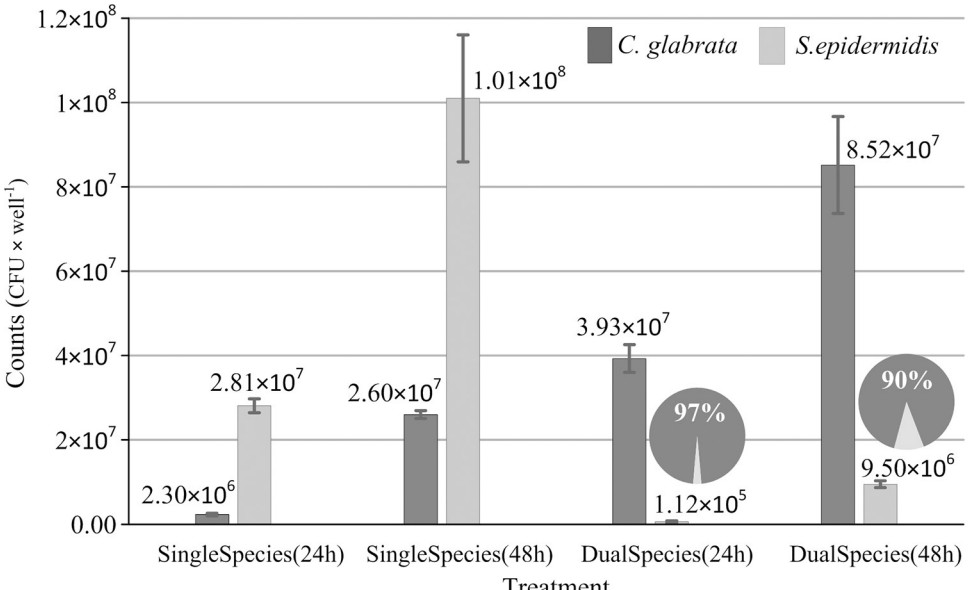

**Fig 2.** Enumeration of colony forming units per well for single- and dual-species biofilms of *C. glabrata* (dark grey bars) and *S. epidermidis* (light grey bars) after 24 h and 48 h incubation. Results are the mean of three replicates and error bars represent 95% confidence limits.

biofilm biomass and cells metabolic activity in dual biofilms are comparable to total biomass and cells metabolic activity measured on single *C. glabrata* biofilms and significantly lower than the corresponding estimates in single *S. epidermidis* biofilms.

To better characterize mono and polymicrobial biofilms, viable cell numbers of both *C. glabrata* and S. *epidermidis* in single and dual-species biofilms, after 24 h and 48 h incubation, were enumerated as colony forming units per well (CFU · well$^{-1}$) and results are exposed in Fig 2.

As expected from Fig 1, the highest counts are recorded for *S. epidermidis* single biofilms (i.e., 2.8×10$^7$ CFU · well$^{-1}$ and 1.0×10$^8$ CFU · well$^{-1}$, respectively, after 24 h and 48 h incubation) and the lowest for *C. glabrata* biofilms (i.e., 2.3×10$^6$ CFU · well$^{-1}$ and 2.6×10$^7$ CFU · well$^{-1}$, respectively, after 24 h and 48 h incubation).

In mixed biofilm, a significant increase of *C. glabrata* and a decrease of *S. epidermidis* counts, with respect to single biofilms of each microorganism, is observed both after 24 and 48 h incubation.

From these values, it can be estimated that mixed biofilms are dominated by *C. glabrata* cells. In fact, after 24 h incubation, *C. glabrata* cells account for about 97% (and *S. epidermidis* for about 3%) of viable cells in the mixed biofilms.

Therefore, it seems that *C. glabrata* CFUs are promoted in dual biofilms. Qualitatively, an analogous effect has been reported comparing *C. albicans* biofilms with *C. albicans/S. mutans* dual biofilms grown in tryptic soy broth containing 0.5% yeast extract (TSBYE) and 1% sucrose. In fact, *C. albicans* CFUs were found to rise by a factor ~10$^{1.5}$ in dual biofilms with respect to pure fungal biofilm (although *S. mutans* was demonstrated to be a stronger biofilm producer than *C. albicans*) [31].

However, from Fig 2 we see that the fraction of *S. epidermidis* cells in mixed biofilms rises to about 10% after 48 h incubation.

To support the above findings from a different perspective, expression of three genes of *C. glabrata* (i.e., *Erg11*, *ALS3* and *FKS1*) and three genes of *S. epidermidis* (i.e., *icaD*, *FnbA* and

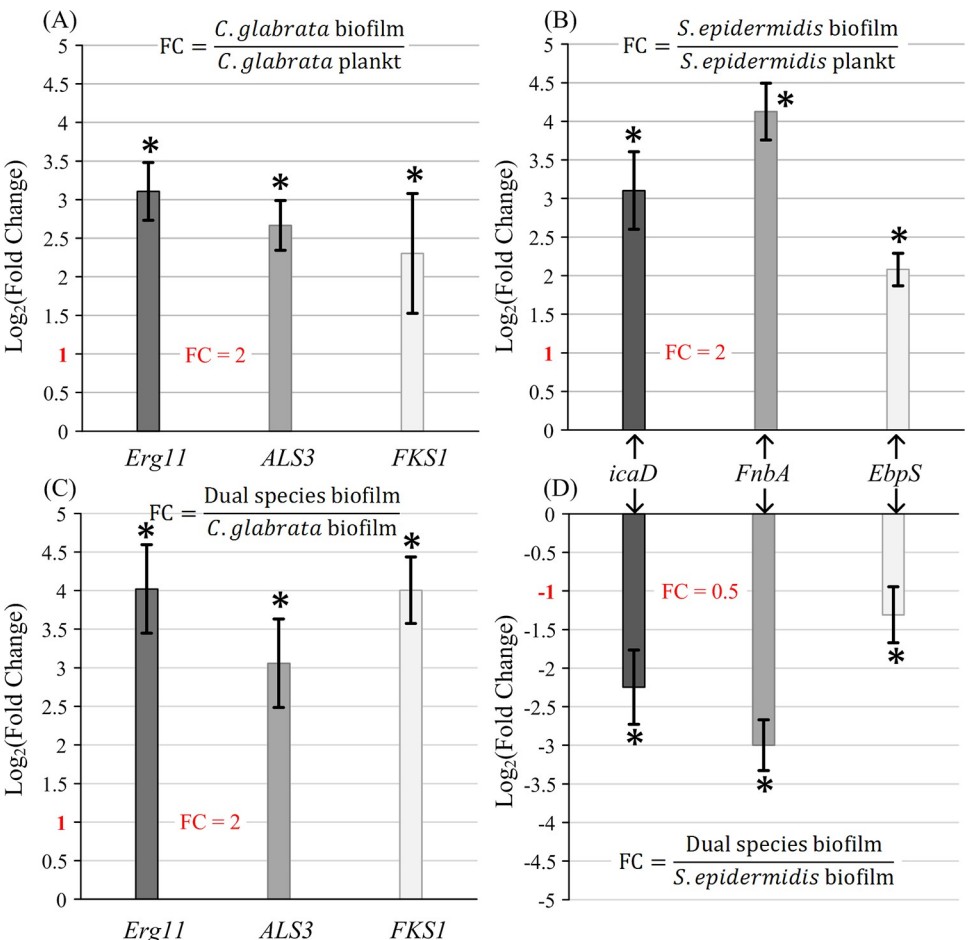

**Fig 3.** Comparison of mRNA expression levels of virulence response genes of *C. glabrata* (*Erg11*, *ALS3* and *FKS1* genes) and *S. epidermidis* (*icaD*, *FnbA*, *EbpS* genes) in planktonic, single-species and dual-species biofilms after 24 h incubation. (A) *C. glabrata* biofilm vs. *C. glabrata* planktonic cells; (B) *S. epidermidis* biofilm vs. *S. epidermidis* planktonic cells; (C) dual-species biofilm vs. *C. glabrata* single-species biofilm; (D) dual-species biofilm vs. *S. epidermidis* single-species biofilm. Data are the mean ± standard deviation of three independent experiments. Asterisks indicate fold changes significantly different from 1 (* = *p*-value < 0.05, *t*-test). FC = Fold Change.

*EbpS*) that are known to be positively related to biofilm production was assessed in planktonic, single-species and dual-species biofilms by qRT-PCR after 24 h incubation (Fig 3).

From Fig 3A and 3B we see that *Erg11*, *ALS3*, *FKS1* genes for *C. glabrata* and *icaD*, *FnbA*, *EbpS* genes for *S. epidermidis* were upregulated in single-species biofilms when compared with the planktonic cells. This confirms the involvement of the considered genes in biofilm formation process. In Fig 3C and 3D expression of the targeted genes for *C. glabrata* and *S. epidermidis* in the dual-species biofilms is compared respectively to their expression in single-species *C. glabrata* and *S. epidermidis* biofilms. With respect to single microbial biofilms, *C. glabrata* genes are upregulated while *S. epidermidis* genes are downregulated which is what one would predict if bacterial growth and biofilm formation were inhibited in the polymicrobial environment.

## Metabolomic footprint analysis

For metabolomic GC-MS footprint analysis, we have acquired 54 observations (GC-MS chromatograms) which span six different cultural conditions or classes (i.e., class#1 = *C. glabrata*

planktonic culture; class#2 = *S. epidermidis* planktonic culture; class#3 = *C. glabrata* biofilm 24 h; class#4 = *S. epidermidis* biofilm 24 h; class#5 = *C. glabrata*/*S. epidermidis* dual biofilm 24 h; class#6 = *C. glabrata*/*S. epidermidis* dual biofilm 48 h). Each class comprised three biological replicates, and GC-MS analysis of each biological replicate was replicated three times, giving nine observations (samples) in each of the six classes.

Each GC-MS acquisition was first presented to the Automated Mass Spectral Deconvolution and Identification System (AMDIS) [21] which result files, segmented according to the six classes, were presented to SpectConnect program [23], which performs correspondence analysis and matches components (metabolites) within and between classes. Furthermore, Spect-Connect creates a relative abundance matrix (RA matrix) with rows corresponding to observations and columns to "conserved metabolites", which is the basis of the following evaluations. A conserved metabolite represents, in the context of SpectConnect software, a signal which consistently persists in technical and biological replicates under identical conditions.

The final dataset comprised 542 chromatographic peaks (components), and, among them, the 106 compounds listed in Table 2 were identified by comparing their deconvoluted 70 eV EI mass spectra with those reported in commercially available libraries and by their Kovats retention index. The last columns in Table 2 contains a 1×6 ordered array of digits defining the coordinates of each identified metabolite which indicate its presence/absence in each of the six cultural classes. This is obtained by associating each condition to a specified number from 1 to 6 (i.e., *C. glabrata* planktonic = 1; *S. epidermidis* planktonic = 2; *C. glabrata* biofilm = 3; *S. epidermidis* biofilm = 4; dual-species biofilm 24 h = 5; dual-species biofilm 48 h = 6). In other words, the six coordinates associated with a metabolite describe its trajectory in the six-dimensional space of classes which can help to establish if a given metabolite is secreted in the medium by *C. glabrata*, S. *epidermidis* or both.

The pie chart in Fig 4 shows that each culture presents almost the same number of conserved identified metabolites and, of the overall number of identified compounds, 50 metabolites are shared among the six classes (i.e., they are consistently found in all cultures and can be identified in Table 2 because they have coordinates {1,2,3,4,5,6}).

The exploration of the data reported in Table 2 reveals the presence in biofilm co-cultures of metabolites exclusively produced by *C. glabrata*. This is the case of 2-phenylethanol, tyrosol and acetyl tyramine (corresponding to coordinates {1,0,3,0,5,6}), of methionol (corresponding to coordinate {1,0,0,0,5,6}) and of xanthine (corresponding to coordinate {0,0,3,0,5,6}) which distribution is presumably related to the presence/absence of *C. glabrata*.

On the opposite side, only 3-phenyllactic acid (coordinates: {0,0,0,4,5,6}), and 2-hydroxyisocaproic acid (coordinates: {0,0,0,4,0,6}) which are detected in the mixed biofilm cultures are presumably produced by *S. epidermidis* (despite the fact that a few metabolites, as for instance acetyl-L-threonine, present coordinates {0,2,0,4,0,0}, which means that they are produced both in planktonic and biofilm pure cultures of *S. epidermidis*).

The prevalence in dual-species biofilm cultures of metabolites characteristic of *C. glabrata*, rather than of *S. epidermidis*, and the absence of some metabolites characteristic of *S. epidermidis* can be explained by the dominance of *C. glabrata* in the mixed biofilms which is also observed by the results showed in Fig 2.

A few metabolites are detected only in dual biofilms because they have coordinate {0,0,0,0,5,6} (e.g., erythritol, which is a very popular sweetener, belongs to this group). All metabolites present in the dual biofilm culture at 24 h are also found after 48 h incubation time. However, in co-cultures after 48 h, additional metabolites are detected, as for instance, tryptophol and *cyclo*-(L-Pro-L-Trp) which have coordinates {0,0,0,0,0,6}.

For multivariate and univariate analysis of GC-MS data described below, the 54×542 RA matrix created by SpectConnect program is imported in our in-house MATLAB m. scripts

**Table 2. Identified extracellular metabolites from metabolomic analysis on six different sample classes: *C. glabrata* planktonic and biofilm 24 h; *S. epidermidis* planktonic and biofilm 24 h; and dual-species *C. glabrata/S. epidermidis* biofilms 24 and 48 h.** RT represents retention time, RI represents Kovats retention index and TMS is the trimethylsilyl function, $(CH_3)_3Si$-. The array of digits in the column labelled "Coordinate" indicates ordinately the class membership of metabolites (*C. glabrata* planktonic = 1; *S. epidermidis* planktonic = 2; *C. glabrata* biofilm = 3; *S. epidermidis* biofilm = 4; dual-species biofilm 24 h = 5; dual-species biofilm 48 h = 6; 0 = not detected).

| ID_## | Name | RT (min) | RI | Coordinate |
|---|---|---|---|---|
| (2) | 2,3-Butanediol, 2TMS | 5.47 | 1057 | {1,2,3,4,5,6} |
| (7) | Lactic Acid, 2TMS | 5.75 | 1077 | {1,2,3,4,5,6} |
| (9) | Glycolic acid, 2TMS | 5.90 | 1087 | {1,2,3,4,5,6} |
| (10) | Pyruvic acid, 2TMS | 6.06 | 1099 | {1,2,3,4,5,6} |
| (12) | Methionol, TMS | 6.19 | 1107 | {1,0,0,0,5,6} |
| (14) | L-Glycine, 2TMS | 6.51 | 1129 | {1,2,3,4,5,6} |
| (16) | 2-Piperidinone, 1TMS | 6.82 | 1152 | {1,2,3,4,5,6} |
| (17) | β-Lactic acid, 2TMS | 6.84 | 1153 | {1,2,3,4,5,6} |
| (19) | 3-Hydroxybutyric acid, 2TMS | 7.07 | 1169 | {1,2,3,4,5,6} |
| (20) | Glyoxylic acid, 2TMS | 7.13 | 1173 | {1,2,3,0,0,0} |
| (22) | 2-Hydroxyvaleric acid, 2TMS | 7.16 | 1175 | {1,2,3,4,5,6} |
| (23) | 2-Aminobutyric acid, 2TMS | 7.26 | 1182 | {1,2,3,4,5,6} |
| (36) | 2-Phenylethanol, TMS | 7.99 | 1234 | {1,0,3,0,5,6} |
| (37) | Urea, 2TMS | 8.15 | 1245 | {1,2,3,0,5,6} |
| (38) | Diethylene glycol, 2TMS | 8.23 | 1251 | {1,2,3,4,5,0} |
| (39) | 4-Methyl 2-keto pentanoic acid enol, 2TMS | 8.31 | 1257 | {1,0,3,0,5,0} |
| (40) | 4,6-Dimethyl-1,3,5-triazin-2-ylamine, TMS | 8.51 | 1271 | {1,2,3,4,5,6} |
| (41) | Isoleucine, 2TMS | 8.95 | 1302 | {1,2,0,0,0,0} |
| (42) | Phosphoric acid, 3TMS | 8.71 | 1285 | {1,2,0,4,5,6} |
| (43) | Nicotinic acid-TMS | 8.89 | 1298 | {1,2,3,4,5,6} |
| (45) | L-Proline, 2TMS | 9.00 | 1306 | {1,2,3,0,5,0} |
| (46) | Succinic acid, 2TMS | 9.17 | 1318 | {1,2,3,4,5,6} |
| (48) | 2,3-Dihydroxy-2-methylpropanoic acid, 3TMS | 9.35 | 1332 | {1,2,3,4,5,6} |
| (49) | Glyceric acid, 3TMS | 9.48 | 1342 | {1,2,3,4,5,6} |
| (50) | Uracil, 2TMS | 9.55 | 1347 | {1,2,3,4,5,6} |
| (51) | 2,5-Dimethyl-4-pyrimidinamine, TMS | 9.57 | 1349 | {1,2,3,4,5,6} |
| (54) | L-Serine, 3TMS | 9.88 | 1372 | {1,2,3,4,5,6} |
| (55) | Pipecolic acid, 2TMS | 9.90 | 1374 | {1,2,3,4,5,6} |
| (60) | *Cyclo*-(L-Ala-L-Gly) | 10.15 | 1392 | {1,2,3,4,5,6} |
| (62) | L-Threonine, 3TMS | 10.24 | 1399 | {1,2,3,4,5,6} |
| (66) | 5-Methylhydantoin, 2TMS | 10.40 | 1412 | {1,2,3,4,5,6} |
| (68) | Trisaminol, 3TMS | 10.53 | 1423 | {1,2,3,4,5,6} |
| (69) | *Cyclo*-(L-Gly-L-Gly) | 10.66 | 1434 | {1,2,3,4,5,6} |
| (71) | Salicyl alcohol, 2TMS | 10.80 | 1445 | {1,2,0,0,0,0} |
| (74) | Homoserine, 3TMS | 11.02 | 1463 | {1,2,3,4,5,6} |
| (76) | 2-Aminomalonic acid, 3TMS | 11.28 | 1484 | {1,2,3,4,5,6} |
| (77) | L-Aspartic acid, 3TMS | 11.31 | 1487 | {1,2,3,4,5,6} |
| (78) | Malic acid, 3TMS | 11.47 | 1500 | {1,2,3,0,0,6} |
| (79) | *Cyclo*-(L-Ornithine) | 11.56 | 1510 | {1,2,3,4,5,6} |
| (81) | Tromethamine, 4TMS | 11.75 | 1532 | {1,2,3,4,5,0} |
| (83) | Pyroglutamic acid, 2TMS | 11.78 | 1535 | {1,2,3,4,5,6} |
| (84) | Hydroxyproline, 3TMS | 11.84 | 1542 | {1,2,3,4,5,6} |
| (85) | 2,5-Piperazinedione, 3-methyl-6-(1-methylpropyl)-, 2TMS | 11.96 | 1556 | {1,2,0,0,0,0} |
| (86) | Tyrosol, 2TMS | 12.18 | 1581 | {1,0,3,0,5,6} |

*(Continued)*

**Table 2.** (*Continued*)

| ID_## | Name | RT (min) | RI | Coordinate |
|---|---|---|---|---|
| (92) | L-Phenylalanine, 2TMS | 12.63 | 1642 | {1,2,3,4,5,6} |
| (94) | Tyramine, 2TMS | 12.66 | 1648 | {1,0,3,0,0,0} |
| (98) | Homoserine, 4-imino-*N,O*-bis(trimethylsilyl)-, trimethylsilyl ester | 12.92 | 1686 | {1,2,3,4,5,6} |
| (102) | *Cyclo*-(L-Gly-L-Pro) | 13.26 | 1745 | {1,2,3,4,5,6} |
| (105) | Glycerol 3-phosphate, 4 TMS | 13.47 | 1783 | {1,2,3,4,5,6} |
| (111) | Myristic acid, TMS | 13.79 | 1850 | {1,2,3,4,5,6} |
| (113) | *Cyclo*-(L-Leu-L-Pro) | 13.91 | 1877 | {1,2,3,4,5,6} |
| (117) | Vitamin B6, 3TMS | 14.09 | 1919 | {1,2,3,4,5,6} |
| (119) | L-Histidine, 3TMS | 14.19 | 1942 | {1,2,3,4,5,6} |
| (121) | Glucitol, 6TMS | 14.28 | 1965 | {1,0,0,0,0,0} |
| (125) | Pantothenic acid, 3TMS | 14.47 | 2013 | {1,2,3,4,5,6} |
| (126) | Palmitic Acid, TMS | 14.58 | 2044 | {1,2,3,4,5,6} |
| (131) | *Cyclo*-(L-Leu-L-Glu), 3TMS | 14.92 | 2136 | {1,0,3,4,5,6} |
| (132) | Oleic Acid, TMS | 15.24 | 2223 | {1,2,3,4,5,6} |
| (133) | Stearic acid, TMS | 15.30 | 2239 | {1,2,3,4,5,6} |
| (134) | L-Tryptophan, 3TMS | 15.34 | 2251 | {1,2,3,4,5,6} |
| (144) | Biotin, 3TMS | 16.46 | 2507 | {1,2,3,4,5,6} |
| (149) | *Cyclo*-(L-Ser-L-Tyr), 2TMS | 17.03 | 2608 | {1,2,3,4,5,6} |
| (151) | Adenosine, 4TMS | 17.40 | 2664 | {1,2,3,4,5,6} |
| (154) | Pyridine, 2-ethyl-5-methyl- | 5.25 | 1043 | {1,2,3,4,5,6} |
| (155) | Ethanolamine, 2TMS | 5.30 | 1046 | {1,2,3,4,5,6} |
| (160) | Thymine, 2TMS | 9.92 | 1375 | {1,2,3,4,5,6} |
| (162) | 3-Aminopiperidine-2,6-dione, 2TMS | 12.46 | 1617 | {1,2,3,4,5,6} |
| (166) | N-Acetyltyramine, 2TMS | 13.88 | 1871 | {1,0,3,0,5,6} |
| (169) | Margaric acid, TMS | 14.94 | 2141 | {1,0,0,0,0,0} |
| (185) | L-Alanine, 2TMS | 8.44 | 1266 | {0,2,3,4,5,6} |
| (191) | 2,5-Dioxypyrazine, 2TMS | 9.79 | 1366 | {0,2,3,4,5,6} |
| (200) | Acetyl-L-threonine, 3TMS | 11.93 | 1552 | {0,2,0,4,0,0} |
| (204) | L-Ornithine, 3TMS | 12.53 | 1627 | {0,0,3,4,5,6} |
| (210) | Xylose, 4TMS | 13.04 | 1704 | {0,2,0,4,0,0} |
| (211) | L-Lysine, 3TMS | 13.12 | 1719 | {0,2,3,4,5,6} |
| (215) | 3-Posphoglyceric acid, 4TMS | 13.70 | 1830 | {0,2,0,0,0,0} |
| (217) | Citric acid, 4TMS | 13.74 | 1840 | {0,2,0,0,0,0} |
| (221) | Adenine, 2TMS | 13.95 | 1886 | {0,2,0,4,5,0} |
| (223) | Talose, 5TMS | 14.10 | 1921 | {0,2,0,0,0,0} |
| (228) | D-Gluconic acid, 6TMS | 14.57 | 2041 | {0,2,0,0,0,0} |
| (231) | Guanine, 3TMS | 14.95 | 2146 | {0,2,0,4,0,0} |
| (236) | Cyclo-(L-Phe-L-Pro) | 15.39 | 2261 | {0,2,3,4,5,6} |
| (245) | Trehalose, 8TMS | 18.54 | 2810 | {0,2,3,4,0,0} |
| (248) | Toluic acid, TMS | 9.83 | 1369 | {0,2,0,0,0,0} |
| (275) | Glycerol, 3TMS | 8.71 | 1285 | {0,0,3,4,0,6} |
| (278) | Fumaric acid, 2TMS | 9.58 | 1350 | {0,0,3,0,0,0} |
| (281) | 2-Deoxytetronic acid, 3TMS | 10.80 | 1445 | {0,0,3,4,5,6} |
| (284) | *Cyclo*-(L-Leu-L-Ala), 2TMS | 11.53 | 1508 | {0,0,3,4,5,6} |
| (293) | L-Glutamic acid, 3TMS | 12.58 | 1635 | {0,0,3,4,5,6} |
| (303) | L-Tyrosine, 2TMS | 14.25 | 1958 | {1,2,3,4,5,6} |
| (304) | Xanthine, 3TMS | 14.57 | 2040 | {0,0,3,0,5,6} |

(*Continued*)

**Table 2.** (Continued)

| ID_## | Name | RT (min) | RI | Coordinate |
|---|---|---|---|---|
| (320) | Cysteine, 3TMS | 12.10 | 1572 | {0,2,3,4,5,6} |
| (340) | 2-Hydroxybutyric acid, 2TMS | 6.63 | 1138 | {0,0,0,4,0,0} |
| (341) | Monoethyl phosphate, 2TMS | 8.01 | 1235 | {0,0,0,4,0,0} |
| (349) | 3-Phenyllactic acid, 2TMS | 12.32 | 1597 | {0,0,0,4,5,6} |
| (350) | L-Asparagine, 4TMS | 12.68 | 1649 | {0,0,3,4,5,6} |
| (399) | Erythritol, 4TMS | 11.64 | 1520 | {0,0,0,0,5,6} |
| (400) | L-Methionine, 2TMS | 11.76 | 1533 | {0,0,0,4,5,6} |
| (412) | Hypoxanthine, 2TMS | 13.66 | 1822 | {0,0,0,0,5,6} |
| (442) | Guanosine, 5TMS | 18.51 | 2807 | {0,0,0,0,5,6} |
| (465) | L-Valine, 2TMS | 7.87 | 1225 | {0,0,0,0,5,6} |
| (466) | 2-Hydroxyisocaproic acid, 2TMS | 8.14 | 1244 | {0,0,0,4,0,6} |
| (518) | Myo-Inositol, 6TMS | 14.89 | 2128 | {0,0,0,0,0,6} |
| (536) | *Cyclo*-(L-Pro-L-Trp) or Brevianamide F | 20.01 | 2948 | {0,0,0,0,0,6} |
| (538) | L-Glutamine, 3TMS | 13.49 | 1788 | {0,0,0,0,0,6} |
| (540) | Tryptophol, 2TMS | 14.08 | 1916 | {0,0,0,0,0,6} |

and reduced to a 54×106 RA matrix of identified metabolites [24]. Before multivariate analysis, to make metabolites equally important, the reduced RA matrix is passed to MATLAB *zscore* function and standardized such that columns have mean 0 and standard deviation 1.

## Multivariate data analysis

**Principal component analysis.** Principal component analysis (PCA) was first carried out to detect intrinsic clustering between samples. PCA scores plot in Fig 5A clearly indicates an unsupervised separation of the six classes in the plane defined by the first and second PCA components which account, respectively, for 41.06% and 18.01% percent of the total predictor's variance. In Fig 5B, it is shown that the 3rd PCA component explains an additional 11.42% of the total variance so that in the tridimensional space defined by the first, second and third PCA components the six classes form well separated and compact groups, except for points representing observations in the *S. epidermidis* biofilm class that expand in the components space due to enhanced biological variability of bacterial biofilm cultures.

**Partial Least Squares Discriminant Analysis (PLS-DA).** It is known that Partial Least Squares Discriminant Analysis (PLS-DA) is a conventional regression approach, which can be applied to categorical response (dependent) variables by encoding them in a so-called dummy variable.

In this approach, $S$ samples (observations) are allocated to $C$ classes and the response dummy variable, **Y**, is a $S \times C$ matrix of 1 and 0 because each class is encoded with one of the $C$ permutations of the $1 \times C$ vector $(1_1 \, 0_2 \, 0_3 \, 0_4. \ldots .0_C)$ so that all observations in the same class correspond to the same permutation.

The dummy variable is then regressed on the predictor's matrix, **X**, which is a numeric matrix of $S$ rows and a number of columns equal to the number of measured predictor variables. An element of the **X** matrix is the value of the predictor variable specified by the column number measured in the observation specified by the row number.

In the present case, the **X** matrix coincides with the 54×106 matrix of identified metabolites relative abundances (RA matrix) created by the SpectConnect program, and the size of the **Y** dummy matrix is 54×6 since our full dataset contains 54 observations allocated to six classes each specified by a permutation of the vector (1 0 0 0 0 0).

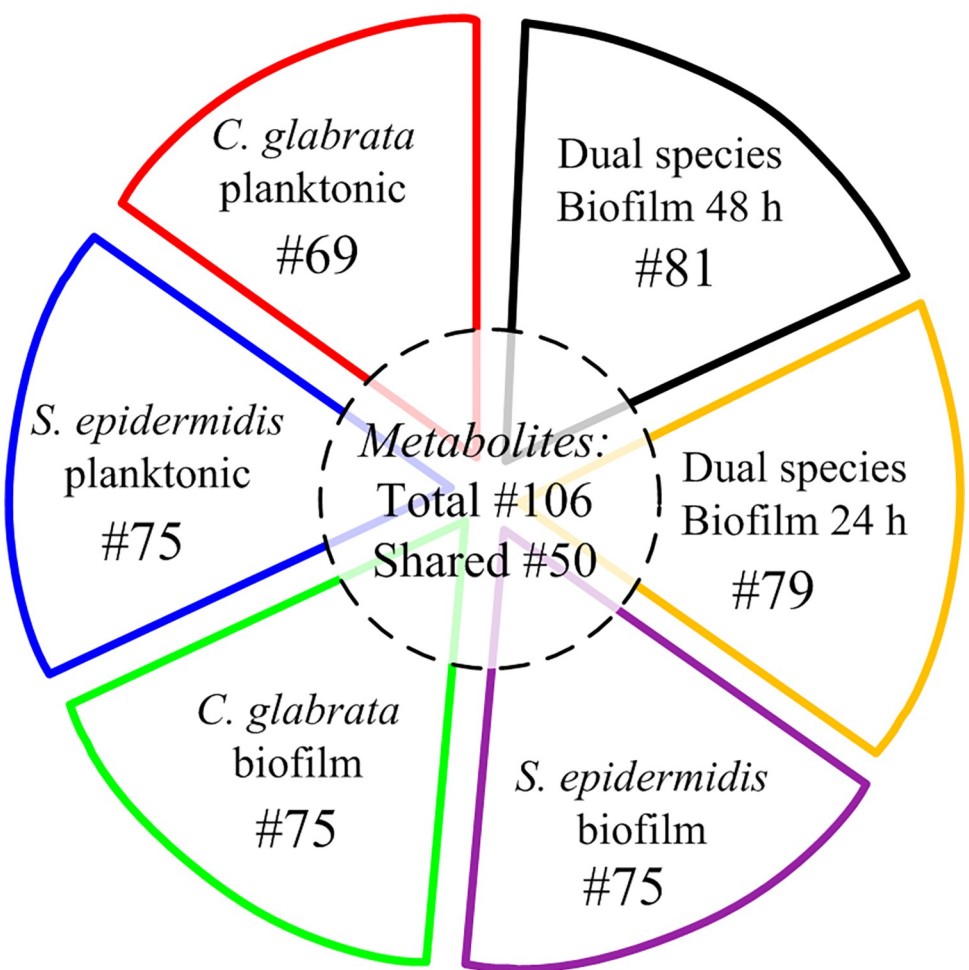

**Fig 4. Number of metabolites identified in cultures of planktonic and biofilm cultures of single- and dual-species of *C. glabrata* and *S. epidermidis* after 24 and 48 h of incubation.**

Metabolomic data are usually high-dimensional, which means that the number of variables (metabolites) is much larger than the number of observations.

PLS-DA overcomes problems connected with high-dimensional data since, analogously to PCA, it is also a dimension reduction technique. In fact, before regression the information contained in the measured variables is extracted and condensed in a much lower number of latent variables or PLS components which are then used for regression in place of the original set of variables.

Dummy-regression based PLS-DA has all the features of a statistical learning technique since a PLS-DA model can be created by analyzing a training dataset made up of predictors with known categorical response (class membership) and the model can be used to predict class membership of unknown observations (test dataset).

A fundamental point to be considered in developing a PLS-DA model is to avoid overfitting the training data. An overfitting model is a model which provides a very close fit of the training set but performs very poorly in predicting class membership of observations in the test dataset and it is obviously useless.

Cross validation (CV) is the most used technique to validate the predictive performance of the PLS-DA model and to avoid overfitting when only the training dataset is available.

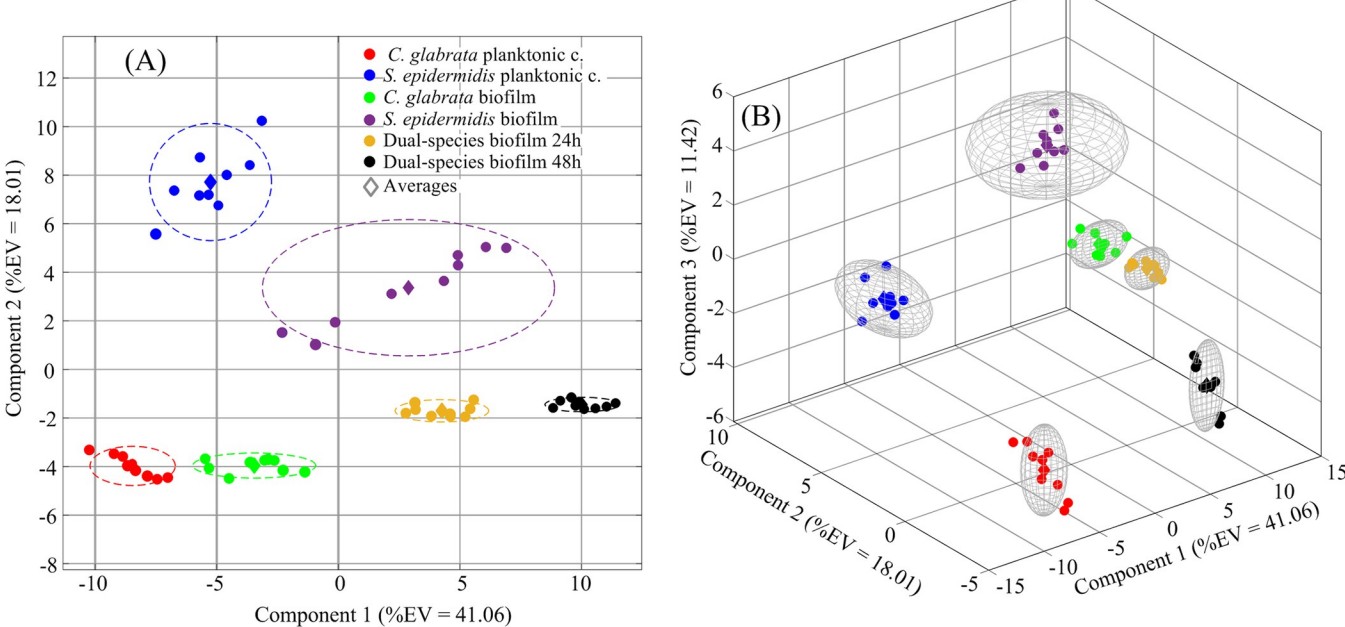

**Fig 5.** (A) 2D and (B) 3D principal component analysis (PCA) scores plots obtained from metabolomic profiles of all six cultural conditions (classes) under examination in this study.

Within the PLS-DA framework, we suggest that cross validation may also be used to check the dependability and strength of a metabolomic dataset. The idea is that if we develop the PLS-DA model on a fraction of observations in a dataset (training set) and then we use this model to predict class membership of remaining observations not used to develop the model (test observations), the error rate in predicting class membership of observations in the test dataset can be interpreted as a measure of the internal consistency of the metabolomic dataset and its general fitness to provide clues for discrimination of different cultural conditions.

To this end, we have developed and applied to our dataset a five-fold CV algorithm based on the scheme in Fig 6A.

The five-fold CV cycle is made up of 5 rounds (from 1 to 5) as described by the column's labels in Fig 6A. In each round, the test set is constituted by an equal number of observations from each of the six classes (rows labels) which constitute a column of the heatmap, and the remaining observations are allocated to the training set. The algorithm selects randomly from the dataset the observations to be collocated in the test set, but it is programmed in such a way that each observation enters the test set only once in the full CV cycle. In other words, no observation is shared between columns of the heatmap.

In each round of the CV cycle, a PLS-DA model is created fitting observations in the training set (42 observations for rounds 1–4 and 48 observations for round 5). Thus, in the CV cycle are created a number of PLS-DA models equal to the number of rounds (i.e., 5).

Each of the five PLS-DA models is applied to the observations in the complementary test set (12 observations for rounds 1–4 and 6 observations for round 5) producing altogether a number of predictions equal to the number of observations in the dataset. Thus, each observation is predicted once on the basis of a model which was developed without using that observation.

All the PLS-DA models in the above CV cycle were developed using 5 PLS components because 89.8% of the Y-variance could be explained with 5 components while 6 components

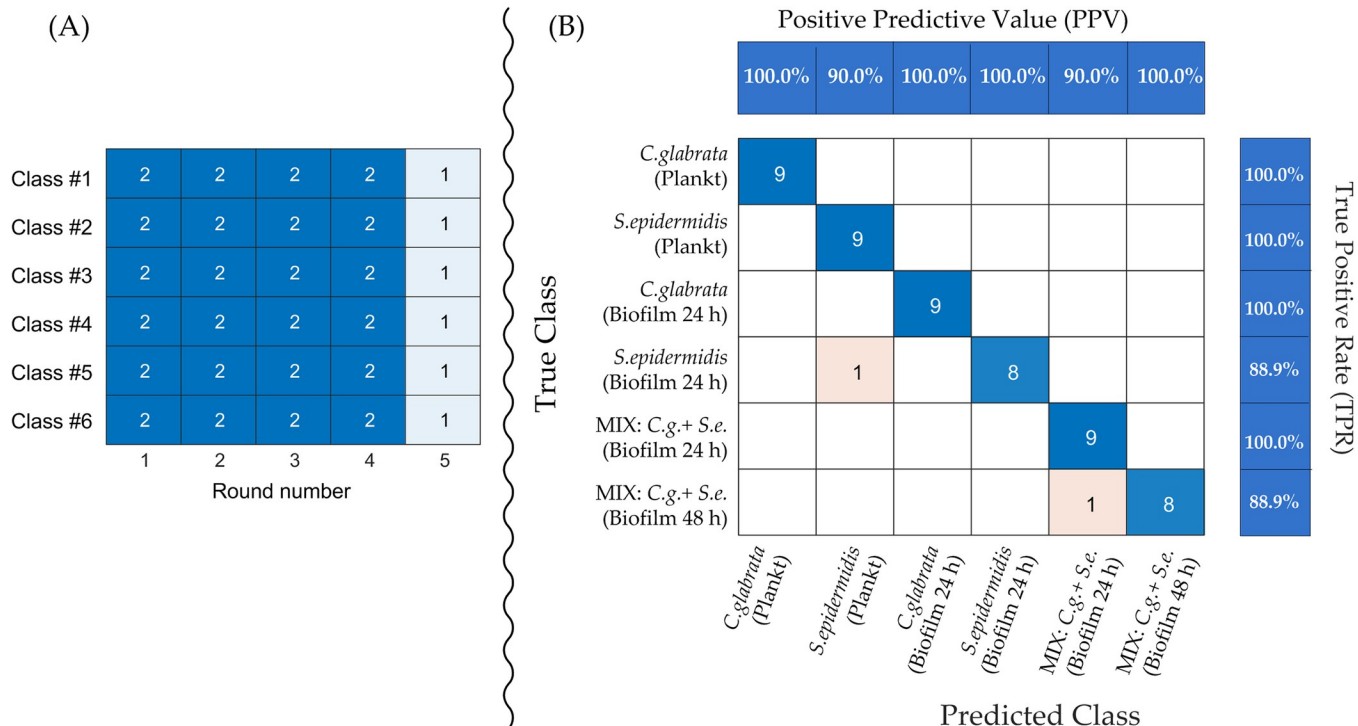

**Fig 6.** (A) Heatmap exposing the five-fold CV cycle employed in this paper to develop a predictive PLS-DA model to classify observations in the dataset collected in this work. (B) Validation confusion matrix resulting from the application of the five-fold cross validation cycle of Fig 6A to the dataset collected in this work. Class#1: *C. glabrata* planktonic culture; Class#2 = *S. epidermidis* planktonic culture; Class#3 = *C. glabrata* biofilm 24 h; Class#4 = *S. epidermidis* biofilm 24 h; Class#5 = *C. glabrata/S. epidermidis* dual biofilm 24 h; Class#6 = *C. glabrata/S. epidermidis* dual biofilm 48 h.

rise this figure to only 93.7% and 4 components sharply lower it to 73.2%. This is a very low number of components for multiclass PLS-DA regression models which generally require a number of components larger than the number of classes and can be interpreted as an indirect indication of the strength of the data [32].

Results of the five-fold CV cycle are conveniently presented as a confusion matrix in Fig 6B and are briefly commented in the following.

From Fig 6B we see that the class of 52 of the 54 observations, which constitute the test set in the above CV cycle, has been correctly predicted. From this, an overall validation accuracy of 96.3% for the PLS-DA model can be evaluated. The True Positive Rate (TPR), which is defined for each class as the percentage of observations of this class which are correctly recognized as members of the class is 100%, except for the *S. epidermidis* biofilm and the co-culture at 48 h for which the TPR drops to 88.9%. The Positive Predictive Value (PPV), which is for each class the percentage of observations attributed to the class which actually belong to the class, falls to 90% only for the planktonic culture of *S. epidermidis* and co-cultures of the microorganisms at 24 h. All this is simply the result of the fact that 1 observation in the *S. epidermidis* biofilm class has been erroneously predicted to belong to the *S. epidermidis* planktonic class and, analogously, one of the observations belonging to the mixed biofilm at 48 h has been misplaced in the mixed biofilm at 24 h class.

Thus, we conclude that an overall error rate of only 3.7% for the validated PLS-DA classifier demonstrates that the metabolomic dataset collected in this work is a coherent source of information on alterations of the metabolism of the studied microorganisms under different cultural conditions.

**Univariate data analysis.** To detect expression changes in metabolites due to the cultural conditions, univariate data analyses were performed to compare metabolites levels, one by one, in pairs of classes (conditions). From the six different conditions examined in this study, in the following we present five pairwise comparisons considered relevant to our purposes.

For each pairwise comparison, Student's *t*-test for equal mean is performed between the two 9x1 vectors of RA values representing each metabolite in the two compared classes. Thus, the statistical univariate analysis associates to each metabolite in the two compared classes, a *p*-value derived from the Student's *t*-test, and a fold change (FC) which is the ratio between the averages of relatives abundances of the metabolite in the two compared classes. At the 5% significance level (i.e., $\alpha = 0.05$), a metabolite is considered differentially expressed in the two classes if the *p*-value is lower than 0.05 and the FC is greater than 2 or lower than 0.5. Results of the 5 pairwise comparisons considered are presented in Figs 7–9 in the form of volcano plots. The ID number exposed in the volcano plots can be used to link differentially expressed metabolites in Figs 7–9 to their coordinates exposed in Table 2.

In the volcano plots of Fig 7, planktonic and biofilm cultures of *C. glabrata* (Fig 7A) and of *S. epidermidis* (Fig 7B) were compared. For both microorganisms, several extracellular metabolites have statistically significant different levels comparing planktonic and biofilm cultures. Among them, several amino acids and several cyclodipeptides (e.g., *cyclo*-(L-Phe-L-Pro), *cyclo*-(L-Leu-L-Ala)) are in the list of extracellular metabolites upregulated in biofilm cultures. Of special significance in Fig 7A is that both glycerol and glycerol 3-phosphate are upregulated in *C. glabrata* biofilm and greatly contribute to the distinction of biofilm from planktonic cultures. The combined relative abundance of glycerol + glycerol 3-phosphate in the *C. glabrata* biofilm is about 14 times larger than in the planktonic culture and, furthermore, it is larger than in any other class. In *C. albicans* a key function has been attributed to the level of glycerol for biofilm formation [33]. In fact, it is reported that the impaired capacity for biofilm formation by a *C. albicans* mutant, in which the gene *RHR2* (encoding the enzyme glycerol-3-phosphatase, acting at the terminal step in glycerol biosynthesis) was non-functional, could fully be restored by supplementing the culture medium with exogenous glycerol. Therefore, it is reasonable to infer that glycerol has a role also for formation and growth of *C. glabrata* biofilm.

From the comparisons in Fig 8 of single- and dual-species biofilm cultures, it can be seen that the metabolomic profile of dual-species biofilm after 24 h of incubation is significantly different from both single biofilm cultures. As can be seen, hypoxanthine (coordinates: {0,0,0,0,5,6}), methionol (coordinates: {1,0,0,0,5,6}), 3-phenyllactic acid (coordinates: {0,0,0,4,5,6}), guanosine (coordinates: {0,0,0,0,5,6}) and 2-phenylethanol (coordinates: {1,0,3,0,5,6}) are upregulated in dual-species biofilm with respect to the single biofilm cultures of both microorganisms.

The comparison of the dual-species biofilm cultures after 24 h and 48 h of incubation shows only a few metabolites with statistically significant different levels (Fig 9). The metabolic similarity is in accordance with the microbial composition similarity observed for the two compared conditions which showed a large, although not identical, prevalence of *C. glabrata* at both incubation periods (Fig 2).

## Discussion

Many chronic infections are attributed to polymicrobial biofilms which frequently exhibit increased resistance to antimicrobial treatments [34]. Despite polymicrobial biofilms represent a clinically relevant health problem, interspecies interactions in mixed infections are understudied and poorly understood [1, 35, 36].

The current study is the first one in literature considering the coexistence of *C. glabrata* and *S. epidermidis* in the polymicrobial biofilm.

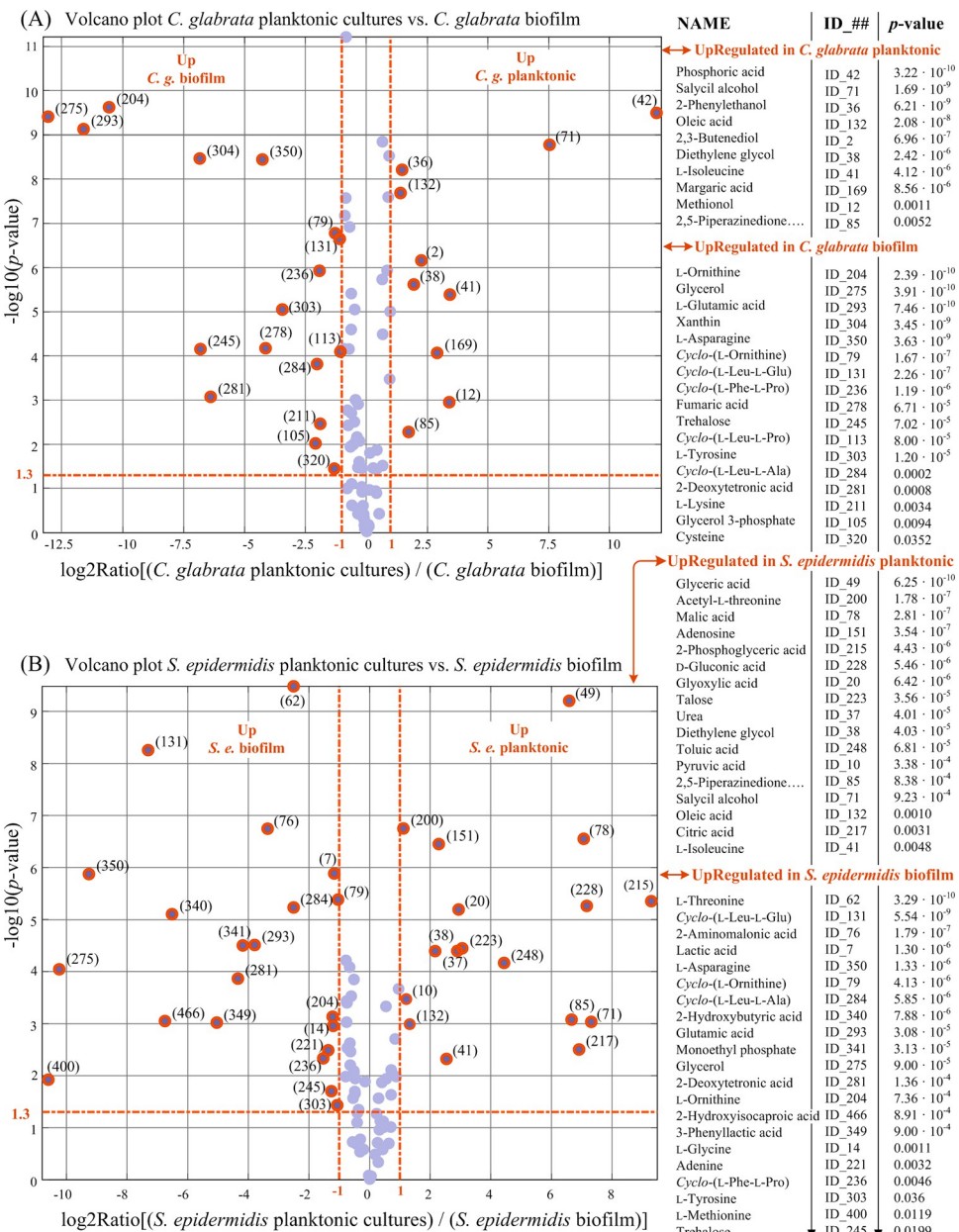

**Fig 7.** Results of univariate comparison of: (A) *C. glabrata* planktonic cultures vs. biofilm cultures; (B) *S. epidermidis* planktonic cultures vs. biofilm cultures. For each metabolite, fold change (FC) is defined as the ratio between mean relative abundances in the compared classes. Only metabolites with *p*-value < 0.05 and fold changes greater than 2 or lower than 0.5 are considered to be differentially expressed in the two compared classes (orange rings). Blue dots represent metabolites with no significantly different levels in the two compared classes. Metabolites can be tracked back to identified metabolites in Table 2 through the associated ID number.

Albeit from a broad perspective interactions between species of *Candida* and *Staphylococcus* are estimated to be synergistic in nature due to metabolic interactions and to physical associations that may facilitate the adherence between the two microorganisms [35, 37], our findings consistently indicate an antagonistic effect of *C. glabrata* DSM11226 towards *S. epidermidis* clinical strain.

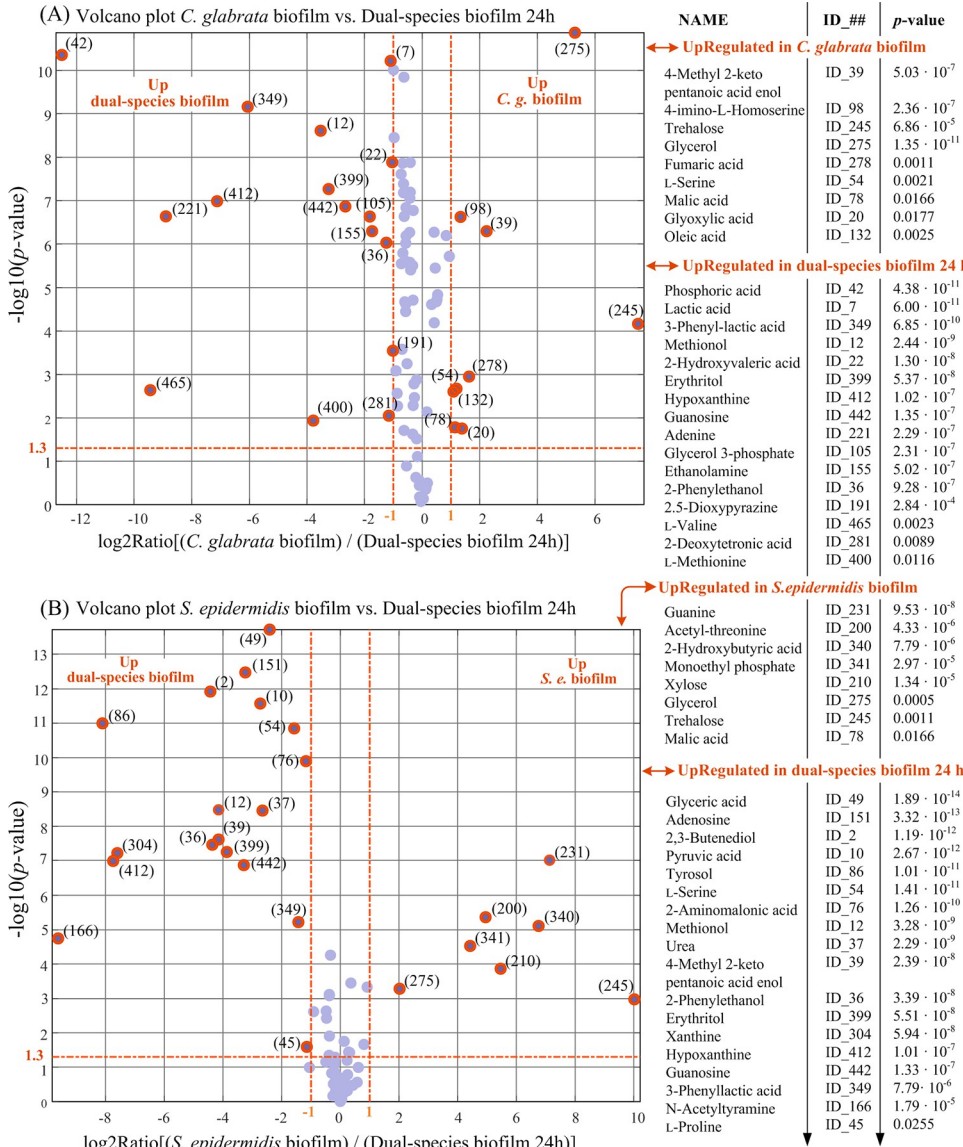

**Fig 8.** Results of univariate comparison of: (A) *C. glabrata* biofilm cultures vs. dual-species biofilm cultures and after 24 h of incubation; (B) *S. epidermidis* biofilm cultures vs. dual-species biofilm cultures after 24 h of incubation. For each metabolite, fold change (FC) is defined as the ratio between mean relative abundances in the compared classes. Only metabolites with *p*-value < 0.05 and fold changes greater than 2 or lower than 0.5 are considered to be differentially expressed in the two compared classes (orange rings). Blue dots represent metabolites with no significantly different levels in the two compared classes. Metabolites can be tracked back to identified metabolites in Table 2 through the associate ID number.

In the first place, the biological data in Figs 1 and 2 point sharply to an inhibition of *S. epidermidis* by *C. glabrata* in mixed *C. glabrata*/*S. epidermidis* biofilms. On the contrary the biofilm producing capacity of *C. glabrata* seems to be stimulated by the presence of *S. epidermidis*. Nevertheless, the moderate increase of the fraction of *S. epidermidis* cells in mixed biofilm from about 3% after 24 h to about 10% after 48 h incubation may be interpreted as the feeble sign of an active competition between the two microorganisms in dual-species biofilms.

Further data on the antagonistic effect of *C. glabrata* toward *S. epidermidis* were obtained from the evaluation of the expression of three selected genes of *C. glabrata* (i.e., *Erg11*, *ALS3*,

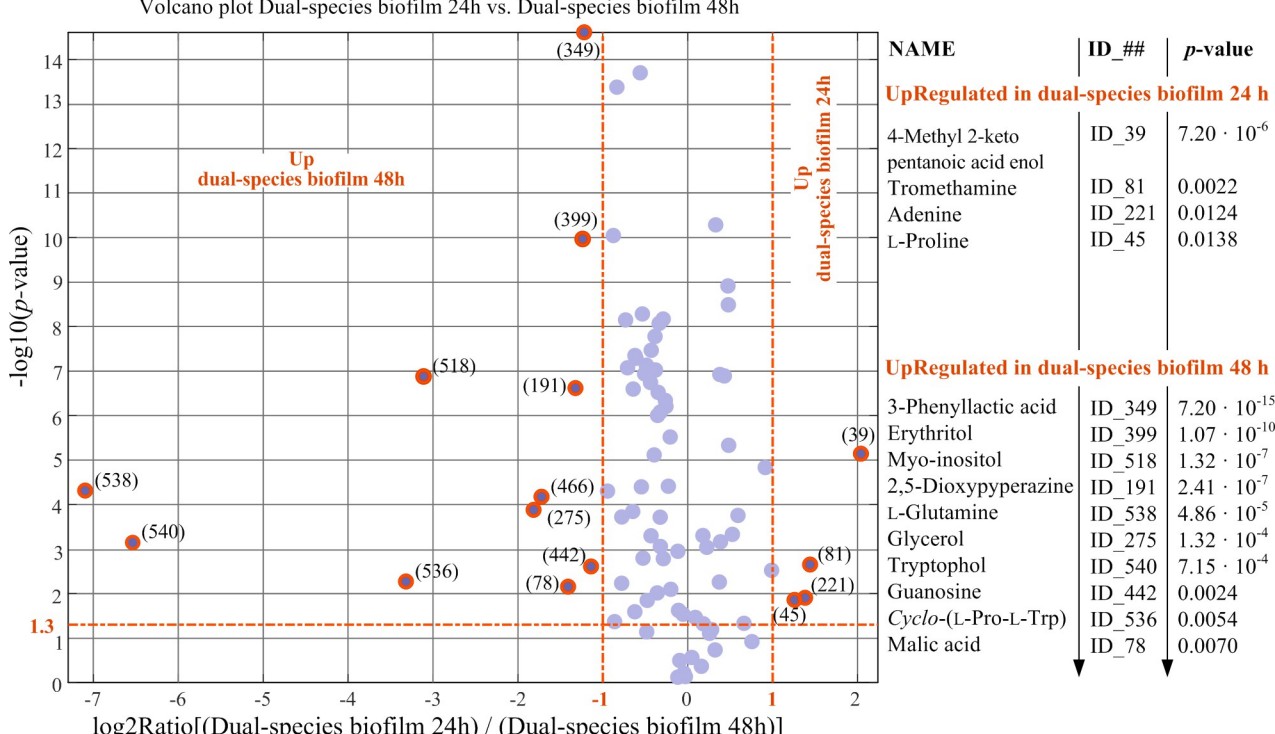

**Fig 9. Results of univariate comparison of dual-species biofilm cultures after 24 h of incubation vs. dual-species biofilm cultures after 48 h of incubation.** For each metabolite, fold change (FC) is defined as the ration between mean relative abundances in dual-species biofilm cultures after 24 h of incubation and in dual-species biofilm cultures after 48 h of incubation. Only metabolites with $p$-value $< 0.05$ and fold changes greater than 2 or lower than 0.5 are considered to be differentially expressed in the two compared classes (orange rings). Blue dots represent metabolites with no significantly different levels in the two compared classes. Metabolites can be tracked back to identified metabolites in Table 2 through the associated ID number.

*FKS1*) and three selected genes of *S. epidermidis* (i.e., *icaD*, *FnbA*, *EbpS*) in single- and dual-species cultures. The regulation of these genes is essential for the microbial adhesion and biofilm formation. In fact, in *Candida* species *ALS3* gene encodes for cell surface proteins that help microorganism to bind to host cells, *Erg11* is a gene involved in the biosynthetic pathway of ergosterol which is a sterol cell membrane component that is unique to fungi and *FKS1* gene encodes the catalytic membrane subunit of glucan synthase which is involved in drug resistance. Analogously, in *Staphylococcus* species, *icaD* gene encodes proteins mediating the synthesis of polysaccharide for intercellular adhesion, *FnbA* encodes for fibronectin-binding proteins A and *EbpS* gene encodes for elastin binding protein.

According to data in Fig 3, the targeted genes appear to be upregulated in biofilm cultures with respect to planktonic cultures, confirming the involvement of the selected genes for biofilm accumulation. On the other side, only the genes involved in the biofilm formation of *C. glabrata* are upregulated in dual-species biofilm when the gene expression of the single-species biofilm cultures is compared to the expression in dual-species biofilm cultures (Fig 3C and 3D). These results strongly suggest an inhibition of the *Staphylococcus* genes transcription during the formation of dual-species biofilms confirming the data obtained from the evaluation of the microbial composition of mixed biofilms which indicate the prevalence of *C. glabrata* on *S. epidermidis*.

Metabolic footprint, which focus on the analysis of metabolites secreted and/or uptaken by cells from the culture medium [38], is a powerful approach to investigate changes in microbial

metabolism under different cultural conditions and it is eminently suitable to track metabolites involved in quorum-sensing, the major mechanism of cell-to-cell communication to control several aspects of microbial life including biofilm formation, morphogenesis, pathogenesis and virulence [39]. Therefore, GC-MS metabolomic data collected in this work allow to throw some light on differences between planktonic and biofilm cultures as well as on metabolic interactions taking place between *C. glabrata* and *S. epidermidis* in dual-species biofilm, on the base of the distribution, through the six culture types considered, of the 106 identified extracellular metabolites that are related to primary and secondary metabolic pathways (see Table 2 and Fig 4).

Detected primary metabolites are essentially amino acids, carbohydrates, dicarboxylic acids and lactic acid.

The tryptone soya broth (TSB) culture medium is a rich medium which contains, between others, an assortment of amino acids which are available for direct uptake by the yeast and bacterial cells. In effect, α-aminobutyric acid, hydroxyproline, pyroglutamic acid, ornithine and all proteinogenic amino acids, except arginine and leucine, have been detected throughout the six cultural conditions examined.

From Fig 7 we see that six amino acids (including, Asn, Cys, Glu, Lys, Orn and Tyr) are upregulated in *C. glabrata* biofilm with respect to planktonic culture. Analogously, seven amino acids (including, Asn, Gly, Glu, Met, Orn, Thr, and Tyr) are upregulated in *S. epidermidis* biofilm with respect to the bacterial planktonic culture. Accumulation of ornithine in the culture medium has been reported during staphylococcal biofilm formation [40, 41]. From our data we see not only that this is true also for *C. glabrata* biofilm but also that upregulation of ornithine in single-species biofilms is twinned with the upregulation of *cyclo*-ornithine (see Fig 7). Furthermore, since amino acids can enter the tricarboxylic acid (TCA) cycle by being metabolized into intermediates of this cycle, reduced consumption of these nutrients imply a reduction of TCA cycle activity in biofilm cultures with respect to planktonic cultures [42]. In *S. epidermidis* and *S. aureus* a reduction of the activity of the TCA cycle has been associated to a high production of polysaccharide intercellular adhesin (PIA) which is an important ingredient in staphylococcal biofilm accumulation [40, 43, 44]. It is reasonable to presume that also for *C. glabrata* the observed reduction of the TCA cycle activity and accumulation of amino acids in the supernatant of biofilm cultures are factors that facilitate biofilm growth.

On the other side, as can be seen from Fig 8, very few amino acids are found to have significantly different levels when single-species and dual-species biofilm cultures are compared. Notably, all biofilm cultures expose not significantly different levels of ornithine and *cyclo*-ornithine.

Trehalose is perhaps the metabolite that most strikingly marks the change in metabolism of the two microorganisms when they are co-cultured. Trehalose is a nonreducing disaccharide of glucose, widespread throughout the biological world, which over the years has been described initially as an energy storage and carbon reserve, then as a stabilizer and protector of membranes and proteins, and, subsequently, as a protector from oxidative and heat stress [45]. From our data, we see that either *C. glabrata* or *S. epidermidis* can produce trehalose which is detected both in the yeast and bacterium biofilms. However, trehalose fades out in the mixed cultures both after 24 h and 48 h incubation. This finding can be rationalized presuming that, in a competitive environment, each microorganism blocks secretion of a metabolite that could be a resource for the other.

Hypoxanthine, a purine-based compound formed during purine catabolism and occasionally found as a constituent of nucleic acids, is another metabolite which has a prominent role in the distinction of *C. glabrata*/*S. epidermidis* biofilms from the pure yeast and bacterial biofilms (see Fig 8). Since hypoxanthine (coordinates: {0,0,0,0,5,6}) manifests itself only in the

dual biofilms, it is not immediately clear if it is produced by *C. glabrata*, *S. epidermidis* or both. However, within a technical metabolomic framework, it can be stated that hypoxanthine is secreted in the medium by *C. glabrata*. In fact, as the name implies, hypoxanthine is the reduced form of xanthine (xanthin + $2H^+$ + $2e^-$ ⇌ hypoxanthine + $H_2O$) which, as can be seen from Table 2, is also an identified metabolite. Technically, xanthine (coordinates: {0,0,3,0,5,6}) is produced by *C. glabrata* since it appears first in the fungal biofilm, and it is a prominent metabolite playing a key role in the discrimination of *C. glabrata* biofilm culture from the planktonic culture (see Fig 7A). Xanthine is not detected in single *S. epidermidis* biofilm, but it is found in dual biofilms at levels comparable with level in the pure *C. glabrata* biofilm, as can be deduced from the fact that it does not appear in the volcano plot of Fig 8A. From this it can be concluded that hypoxanthine is produced by *C. glabrata* both in the single and in the mixed biofilms and the fact that hypoxanthine manifests itself only in the mixed biofilms is simply because in the single *C. glabrata* biofilm it is completely oxidized to xanthine.

In this study, a result that should be emphasized concerns the detection in the investigated cultures of an assortment of secondary metabolites (e.g., aromatic and non-aromatic alcohols and amines, cyclodipeptides, hydroxy and non-proteinogenic amino acids, pyrazine and pyridine derivatives, etc.) which contribute to the discrimination between cultural conditions. A bunch of these secondary metabolites are known to perform quorum-sensing functions [39, 46]. In fact, quorum-sensing molecules are produced by both bacteria and fungi to govern a variety of behaviors.

Tyrosol is a well-known aromatic alcohol which is produced by several fungi. We find that tyrosol is also consistently produced by *C. glabrata* because it is detected in all cultures in which the fungus is present (as can be deduced from its coordinates in Table 2: {1,0,3,0,5,6}). Furthermore, tyramine and its acetylated product (N-acetyl tyramine) are scattered through cultures in which *C. glabrata* is present, consistently with the fact that the first step in the biosynthesis of tyrosol from tyrosine is a decarboxylation reaction which converts tyrosine to tyramine [47]. However, tyrosol does not appear in the volcano plots of Figs 7A and 8A because it has not significantly different levels in supernatants of single *C. glabrata* cultures and mixed biofilms. It is reported that in *C. albicans* tyrosol acts as a quorum-sensing molecule for biofilms as well as for planktonic cells and its function is to accelerate conversion of yeast in hyphae [39]. However, since *C. glabrata* only grows in the yeast form and it is lacking a morphological transition from yeast to hyphae, tyrosol appears only to be necessary for regulating biofilm initiation and/or development [39]. This is in agreement with the observed production of tyrosol by *Candida auris* which, as *C. glabrata*, grows as yeast and it is capable of developing biofilms [48].

*Candida* species are also reported to yield tryptophol, a quorum-sensing indolyl alcohol which is produced from tryptophan [46]. It is remarkable that tryptophol biosynthesis from tryptophan involves the same three steps which are performed for the biosynthesis of tyrosol from tyrosine. This consists first in a decarboxylation reaction, which, from tyrosine or tryptophan, produces respectively tyramine or tryptamine; second, the amine carbon of tyramine or tryptamine is oxidized to yield an aldehydic group; finally, the aldehydic group is reduced to produce the alcoholic function respectively of tyrosol or tryptophol. Nevertheless, even though tryptophan is widespread through all cultures examined, tryptophol is exclusively detected in the metabolic profiles of dual-species biofilms after 48 h incubation (see Fig 9), and this suggests that quorum-sensing functions of tyrosol and tryptophol may be different. In this regard it can be mentioned that an autoantibiotic activity towards *C. albicans* has been attributed to tryptophol, which is associated and strictly related to its role as a quorum-sensing molecule [49].

In *C. albicans*, *Candida dubliniensis* and *C. auris* also 2-phenylethanol, which is structurally related to tyrosol, is reported to perform quorum sensing functions and, among other

functions, it is also considered an hyphae-inhibiting metabolite [46, 48]. We find that tyrosol and 2-phenylethanol (coordinates: {1,0,3,0,5,6}) span the same trajectory in the space of classes with a fundamental difference: 2-phenylethanol is upregulated in dual cultures and participates to the distinction between pure *C. glabrata* and *C. glabrata/S. epidermidis* biofilms (see Fig 8A).

3-Phenyllactic acid is structurally related to 2-phenylethanol (in abstract, 2-phenylethanol can be produced by decarboxylation of 3-phenyllactic acid) and both molecules are known to possess broad spectrum antimicrobial activity against bacteria and fungi [50, 51]. However, the trajectory of 3-phenyllactic acid (coordinates: {0,0,0,4,5,6}) in the space of classes is different because 3-phenyllactic acid appears to be produced by *S. epidermidis* during biofilm formation and it is upregulated in mixed cultures with respect to *S. epidermidis* biofilm (see Fig 8B). Thus, in a symmetrical manner, increased secretion of 3-phenyllactic acid by *S. epidermidis* in mixed cultures is the response of the bacterium for the presence of the fungus while increased production of 2-phenylethanol by *C. glabrata* represents the response of the fungus for the presence of the bacterium. As we will see below, the biological significance of these facts is most probably connected with the antimicrobial activity of 3-phenyllactic acid and 2-phenylethanol [51].

In addition, a few low molecular weight nonaromatic alcohols, such as methionol and erythritol, are detected in the mixed cultures and have prominent roles in the distinction of pure biofilms from mixed biofilms (see Fig 8). However, their function is not understood, although methionol is also produced by *C. auris* [48].

Nine cyclodipeptides were detected in the examined cultures. Cyclodipeptides, also known as diketopiperazines, are secondary metabolites obtained from the condensation of two amino acids and are distributed in many organisms including fungi, bacteria, plants and animals where they perform a variety of biologically useful functions [46]. The production of these small molecules seems to be also relevant for its role in the cell-to-cell communication via quorum-sensing [52, 53].

From our data, it results that eight of the nine cyclodipeptides detected can be produced either by *C. glabrata* or *S. epidermidis* since they are detected both in the single biofilm and in the mixed biofilm cultures of the two microorganisms at comparable levels (this is the reason because they do not appear in the volcano plots of Fig 8). However, as can be seen from Fig 7, a number of cyclodipeptides are upregulated in the biofilms with respect to planktonic cultures suggesting that they are secondary metabolites important for biofilm formation and growth. The simultaneous accumulation in biofilm cultures of amino acids and cyclodipeptides is probably not unrelated. For instance, as we have noted above, leucine has never been detected in any of the examined cultures but two dicyclopeptides containing leucine (i.e., *cyclo*-(L-Leu-L-Ala), *cyclo*-(L-Leu-L-Glu)) are invariably observed in biofilm cultures and, furthermore, *cyclo*-(L-Leu-L-Pro) is also detected in planktonic cultures.

The exception to this scenario is constituted by *cyclo*-(L-Pro-L-Trp), also known as Brevianamide F, which is only detected in the co-culture after 48 h incubation and has a role in the distinction of the mixed biofilm at 24 h from the mixed biofilm at 48 h (see Fig 9). It is known that *cyclo*-(L-Pro-L-Trp) possesses antifungal activity against *C. albicans* and *C. tropicalis* and its activity against *C. albicans* is better than amphotericin B, the standard antifungal agent [54]. From Fig 9, we see that *cyclo*-(L-Pro-L-Trp) manifests itself, in the mixed culture at 48 h, simultaneously with an increased secretion of 3-phenyllactic acid, which, according to the above discussion, is presumably produced by *S. epidermidis* in response to the presence of the fungus competitor. From this, we may infer that also *cyclo*-(L-Pro-L-Trp) is most likely produced by *S. epidermidis* to reinforce the antifungal action of 3-phenyllactic acid. In fact, it has been documented that combining 3-phenyllactic acid with the antifungal *cyclo*-(L-Pro-L-Phe), which, as

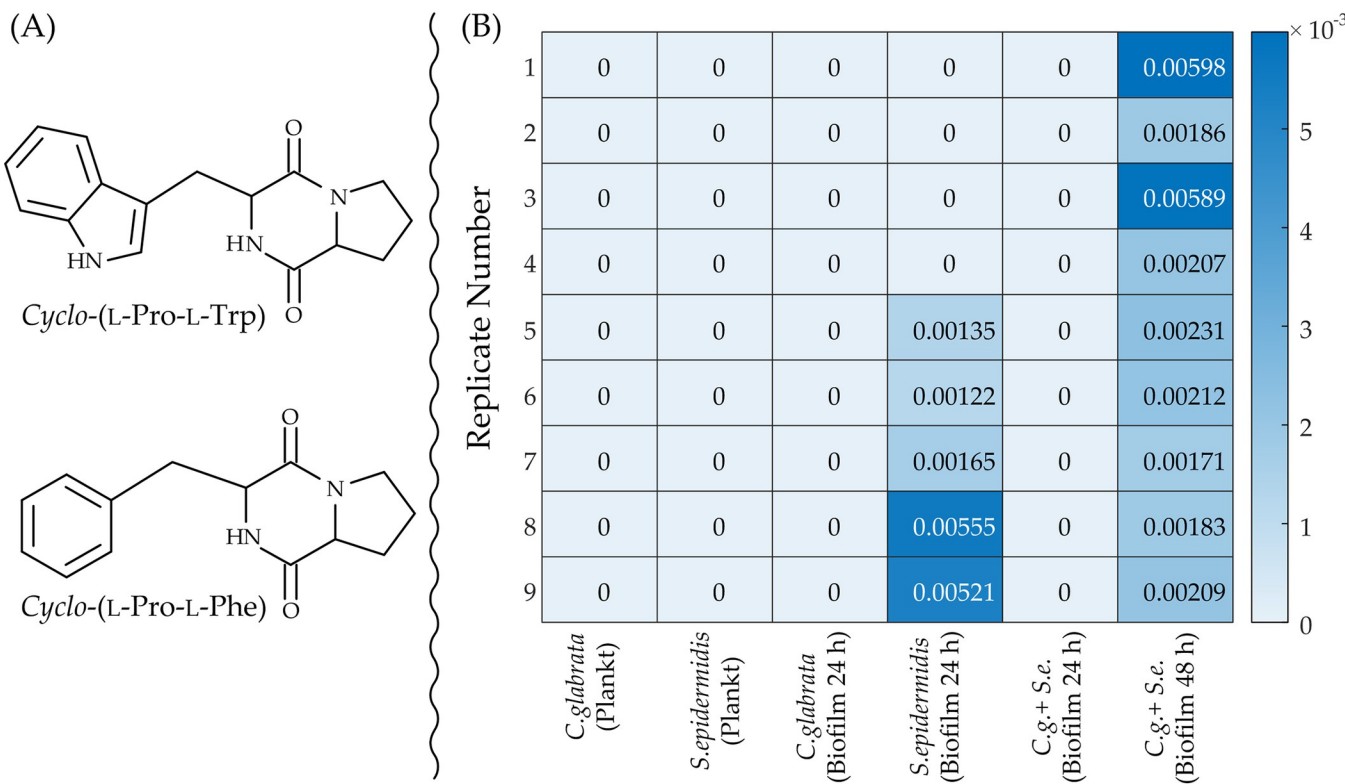

**Fig 10.** (A) Comparison of structures of *cyclo*-(ʟ-Pro-ʟ-Trp) and *cyclo*-(ʟ-Pro-ʟ-Phe). (B) Relative abundances of the antifungal cyclodipeptide *cyclo*-(ʟ-Pro-ʟ-Trp) extracted from the RA matrix created by the SpectConnect program. Each cell of the matrix contains the relative abundance of *cyclo*-(ʟ-Pro-ʟ-Trp) in the class identified by the column label measured in the replicate identified by the row label. *Cyclo*-(ʟ-Pro-ʟ-Trp) has been detected only in 55.5% of replicates in the *S. epidermidis* biofilm class and in 100% of replicates in the dual-species biofilm at 48 h.

can be seen from Fig 10A, is structurally similar to *cyclo*-(ʟ-Pro-ʟ-Trp), leads to a synergistic antifungal effect [55]. As a result, the synergistically combined activity of *cyclo*-(ʟ-Pro-ʟ-Trp) and 3-phenyllactic acid against *C. glabrata* may be the determining factor of the fact, exposed in Fig 2, that the fraction of *S. epidermidis* cells in the mixed biofilm rises from 3% at 24 h to 10% at 48 h. Additionally, Fig 10B, which is an excerpt of the matrix of relative abundances (RA matrix) created by SpectConnect program, shows the relative abundances of *cyclo*-(ʟ-Pro-ʟ-Trp) throughout replicates of the six cultural conditions considered and proves that *S. epidermidis* can produce this key cyclodipeptide. In fact, we see from Fig 10B that, beyond being consistently detected in the dual-species biofilm at 48 h, *cyclo*-(ʟ-Pro-ʟ-Trp) is only detected in 55.6% of replicates in the *S. epidermidis* biofilm class and it is never detected in the other four cultural conditions. A frequency through replicates in the same class of 55.6% is below the threshold frequency of 75% which the SpectConnect program applies by default for assigning to a component the status of "conserved metabolite" in the class. This is because, according to its {0, 0, 0, 0, 0, 6} coordinates, *cyclo*-(ʟ-Pro-ʟ-Trp) trajectory in the space of classes does not include *S. epidermidis* biofilm class. Nevertheless, from Fig 10B there can be no doubt that *cyclo*-(ʟ-Pro-ʟ-Trp) is secreted by *S. epidermidis*.

## Conclusions

Biological assays (including, safranin-O staining/de-staining assay for quantification of biofilm biomass, XTT reduction assay for testing biofilm metabolic activity, and enumeration of

colony forming units for *C. glabrata* and *S. epidermidis*) along with genes expression analysis by qRT-PCR (targeted to *C. glabrata* genes *Erg11*, *ALS3* and *FKS1* and to *S. epidermidis* genes *icaD*, *FnbA*, *EbpS*) performed on single- and dual-species biofilms demonstrate that *C. glabrata*/*S. epidermidis* dual-species biofilm is dominated by yeast cells.

GC-MS metabolomic analysis of metabolites secreted in the culture medium by the two microorganisms provides a molecular view of metabolic events which characterize planktonic and single-species biofilms as well as dual-species biofilms. PLS-DA analysis of the broad metabolomic dataset collected in this study demonstrates that primary and secondary microbial metabolism, in planktonic and in single-species biofilms of each microorganism as well as in dual-species biofilms, are markedly different so that the six different cultural conditions considered in this study can be readily differentiated based on extracellular metabolites. In fact, the PLS-DA classifier developed in this study and validated by a five-fold cross validation algorithm implemented in MATLAB R2021a, is capable of assigning observations to the class they belong with an accuracy of 96.3%.

In all biofilm cultures, an assortment of cyclic dipeptides, which can be produced either by the yeast or the bacterium, is detected. However, the cyclic dipeptide *cyclo*-(L-Pro-L-Trp), which is known to possess antifungal activity, is very likely produced by *S. epidermidis* and secreted in the medium along with 3-phenyllactic acid seems to be capable of mitigating the prevalence of *C. glabrata* in dual-species biofilms.

Although the specific object of this study was to broaden general knowledge on the dual-species fungus/bacterium biofilm formation and growth, we have, for technical reasons related to the interpretation of data collected on dual-species biofilm cultures, extended the investigation to single *C. glabrata* and single *S. epidermidis* planktonic and biofilm cultures. Therefore, the collected biological and metabolomic data may be of wide general interest to the field of microbial infections treatment and prevention. However, any model is to some extent contingent and dependent on the data and conditions under which it was developed, and further research is undoubtedly needed to strengthen our understanding and to comprehend the effects of factors such as specific strains of microorganisms, the biochemical environment, and the nature of the surface on which biofilms are deposited.

## Author Contributions

**Conceptualization:** Maria Michela Salvatore, Anna Andolfi, Francesco Salvatore, Marco Guida, Emilia Galdiero.

**Formal analysis:** Maria Michela Salvatore, Angela Maione, Alessandra La Pietra, Federica Carraturo, Alessia Staropoli.

**Methodology:** Maria Michela Salvatore, Angela Maione.

**Writing – original draft:** Maria Michela Salvatore, Emilia Galdiero.

**Writing – review & editing:** Maria Michela Salvatore, Francesco Vinale, Anna Andolfi, Francesco Salvatore, Marco Guida, Emilia Galdiero.

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
