## [Decision Letter · Decision Letter 0]

14 Oct 2022

PONE-D-22-20604Microbial Interactions and Metabolomic Alterations in Candida glabrata-Staphylococcus epidermidis Dual-Species BiofilmsPLOS ONE

Dear Dr. Galdiero,

Thank you for submitting your manuscript to PLOS ONE. After careful consideration, we feel that it has merit but does not fully meet PLOS ONE’s publication criteria as it currently stands. Therefore, we invite you to submit a revised version of the manuscript that addresses the points raised during the review process.

We are sorry it has taken so long to respond. There has been difficulty getting reviewers and several researchers as well as a proposed reviewer felt that the manuscript was asking a question about an interrelationship that does not physiologically exist. Additionally it was felt that C. glabrata is not a good biofilm maker. Indeed your studies really address dual species interaction and not necessarily in a biofilm. Consequently in the discussion you might include a few sentences on how your data would the interrelationship between the two species even if biofilms werent formed and why this would be important.  

Reviewer 1 asks for more samples but I realize these experiments are usually only done with one strain - so you should comment in discussion that these results are for only 1 strain and differences may exist since it is known that the ability to make biofilms can be strain dependent. 

We look forward to receiving your revised manuscript.

Kind regards,

Joy Sturtevant

Academic Editor

PLOS ONE

Journal Requirements:

“NO authors have competing interests”

Reviewers' comments:

Reviewer's Responses to Questions

**Comments to the Author**

1. Is the manuscript technically sound, and do the data support the conclusions?

Reviewer #1: Yes

2. Has the statistical analysis been performed appropriately and rigorously? 

Reviewer #1: Yes

3. Have the authors made all data underlying the findings in their manuscript fully available?

Reviewer #1: Yes

4. Is the manuscript presented in an intelligible fashion and written in standard English?

Reviewer #1: Yes

5. Review Comments to the Author

Reviewer #1: This article entitled " Microbial Interactions and Metabolomic Alterations in Candida glabrata-Staphylococcus epidermidis Dual-Species Biofilms " is interesting and presents excellent results. The aim of the study is appropriate and novelty in polymicrobial infections field. The study is carried out in a very exhaustive way, both in the application of various techniques for the characterization of polymicrobial biofilms and in their discussion. This is an interesting, comprehensive and well-designed document, and I recommend it for publication. However, I have some considerations to make:

Comments/Questions:

- The sample number is low, would it be possible to increase the number of isolates? Why, use the authors a clinical isolate of Staphylococcus epidermidis and a reference strain of Candida glabrata? More strains of C. glabrata and S. epidermidis should be used!

- Line 182: "Sample preparation", the first two paragraphs are repetitive

- I think the data in table 2 would be more easily perceptible in graph.

- The quality of the imagens in figures should be improved.

6. PLOS authors have the option to publish the peer review history of their article (what does this mean?). If published, this will include your full peer review and any attached files.

Reviewer #1: No

---

## [Author Response · Author response to Decision Letter 0]

21 Oct 2022

Editor: We are sorry it has taken so long to respond. There has been difficulty getting reviewers and several researchers as well as a proposed reviewer felt that the manuscript was asking a question about an interrelationship that does not physiologically exist. Additionally it was felt that C. glabrata is not a good biofilm maker. Indeed your studies really address dual species interaction and not necessarily in a biofilm. Consequently in the discussion you might include a few sentences on how your data would the interrelationship between the two species even if biofilms werent formed and why this would be important. 

Author response: We wish to thank the academic editor for the comments and useful suggestions.

Editor: Reviewer 1 asks for more samples but I realize these experiments are usually only done with one strain - so you should comment in discussion that these results are for only 1 strain and differences may exist since it is known that the ability to make biofilms can be strain dependent.

Author response: According to the academic editor suggestion, text were added in the conclusion section to better explain the focus and relevance of our study which could be of wide general interest to the field of polymicrobial infections. Furthermore, as required by the editor, we also explicitly specified the strain dependence of our findings (see lines 835-845). We have added text in the conclusion rather than in the discussion since we consider extension of our approach to the investigation of polymicrobial biofilm formation and growth to other strains and microbial species to be the object of future research.

Reviewer 1: This article entitled " Microbial Interactions and Metabolomic Alterations in Candida glabrata-Staphylococcus epidermidis Dual-Species Biofilms " is interesting and presents excellent results. The aim of the study is appropriate and novelty in polymicrobial infections field. The study is carried out in a very exhaustive way, both in the application of various techniques for the characterization of polymicrobial biofilms and in their discussion. This is an interesting, comprehensive and well-designed document, and I recommend it for publication. 

Author response: We wish to thank the reviewer for the positive comments and useful suggestions to improve the manuscript.

Reviewer 1: However, I have some considerations to make:

Comments/Questions:

- The sample number is low, would it be possible to increase the number of isolates? Why, use the authors a clinical isolate of Staphylococcus epidermidis and a reference strain of Candida glabrata? More strains of C. glabrata and S. epidermidis should be used!

Author response: The focus of this manuscript is to understand if there was an interaction at biological and metabolomic level between C. glabrata and S. epidermidis in the polymicrobial environment. To achieve this goal we had, by necessity rather than by choice, to select a specified strain of each microorganism since, as all microbiological results, our findings may be strain dependent. This is in fact the reason why in Materials and Methods there is a precise information about the strains employed. In the revised version of the manuscript, we explicitly mentioned in the conclusion this aspect of our data and results. Comparing biological and metabolic characteristics of several strains of C. glabrata and S. epidermidis is outside of the scope of this paper although we consider it an excellent idea for future research. 

Reviewer 1: Line 182: "Sample preparation", the first two paragraphs are repetitive

Author response: As suggested by the reviewer, in the revised version of the manuscript a few sentences were merged to avoid repetitions and, furthermore, the paragraphs “Sample preparation” and “GC-MS analysis” were combined in only one paragraph named “Sample preparation and GC-MS analysis” in which the essential information concerning sample preparation, derivatization before GC-MS analysis, the instrumental method and the number of replicates were reported (see lines 183-206) .

Reviewer 1: I think the data in table 2 would be more easily perceptible in graph.

Author response: As suggested by the reviewer, in the revised version of the manuscript Table 2 was replaced with a figure which was numbered as Figure 2 and, therefore, all subsequent figures and tables were renumbered. 

Reviewer 1: The quality of the imagens in figures should be improved.

Author response: As suggested by the reviewer, the quality and general aspect of all the figures were scrutinized and their quality greatly improved. A few images were restyled to make their content clearer. All images are provided in .tif format 300×300 pixel/in

---

## [Decision Letter · Decision Letter 1]

10 Nov 2022

PONE-D-22-20604R1Microbial Interactions and Metabolomic Alterations in Candida glabrata-Staphylococcus epidermidis Dual-Species BiofilmsPLOS ONE

Dear Dr. Galdiero,

Thank you for submitting your manuscript to PLOS ONE. After careful consideration, we feel that it has merit but does not fully meet PLOS ONE’s publication criteria as it currently stands. Therefore, we invite you to submit a revised version of the manuscript that addresses the points raised during the review process.

Authors should take into account the reviewers' observations, especially in relation to the conclusion drawn from experiments carried out only in microbiological media.

We look forward to receiving your revised manuscript.

Kind regards,

Marcos Pileggi, Ph.D

Academic Editor

PLOS ONE

Reviewers' comments:

Reviewer's Responses to Questions

**Comments to the Author**

1. If the authors have adequately addressed your comments raised in a previous round of review and you feel that this manuscript is now acceptable for publication, you may indicate that here to bypass the “Comments to the Author” section, enter your conflict of interest statement in the “Confidential to Editor” section, and submit your "Accept" recommendation.

Reviewer #1: All comments have been addressed

Reviewer #2: (No Response)

2. Is the manuscript technically sound, and do the data support the conclusions?

Reviewer #1: Yes

Reviewer #2: Partly

3. Has the statistical analysis been performed appropriately and rigorously? 

Reviewer #1: Yes

Reviewer #2: Yes

4. Have the authors made all data underlying the findings in their manuscript fully available?

Reviewer #1: Yes

Reviewer #2: Yes

5. Is the manuscript presented in an intelligible fashion and written in standard English?

Reviewer #1: Yes

Reviewer #2: Yes

6. Review Comments to the Author

Reviewer #1: (No Response)

Reviewer #2: The main drawback of this manuscript is about the conclusion drawn from the experiments carried out on just microbiological media. The authours claim that the choice of the strains are do to its prevalence in skin infections. So I strongly suggest that the metabolome study and gene regulation studies should be carried out using a simulated environment like a wound medium or a simulated wound infection model

7. PLOS authors have the option to publish the peer review history of their article (what does this mean?). If published, this will include your full peer review and any attached files.

Reviewer #1: No

Reviewer #2: No

---

## [Author Response · Author response to Decision Letter 1]

15 Nov 2022

Editor: Authors should take into account the reviewers' observations, especially in relation to the conclusion drawn from experiments carried out only in microbiological media.

Answer: We thank the academic editor for considering our paper. We have carefully considered the Reviewer#2 observations concerning the use of a standard medium for growing single and dual species biofilms of C.glabrata/S. epidermidis biofilms. You can see our reply to this question in the Cover Letter and below in the answer to Reviewer#2. 

Reviewer#1: “This article entitled " Microbial Interactions and Metabolomic Alterations in Candida glabrata-Staphylococcus epidermidis Dual-Species Biofilms " is interesting and presents excellent results. The aim of the study is appropriate and novelty in polymicrobial infections field. The study is carried out in a very exhaustive way, both in the application of various techniques for the characterization of polymicrobial biofilms and in their discussion. This is an interesting, comprehensive and well-designed document, and I recommend it for publication.” 

Answer: We thank the Reviewer for the positive comments.

 Reviewer#2: “The main drawback of this manuscript is about the conclusion drawn from the experiments carried out on just microbiological media. The authours claim that the choice of the strains are do to its prevalence in skin infections. So I strongly suggest that the metabolome study and gene regulation studies should be carried out using a simulated environment like a wound medium or a simulated wound infection model”

Answer: We thank the Reviewer for his/her observation which we have carefully considered. 

We understand that the main concern of the Reviewer is on the use of a standard microbiological medium (TSB medium supplemented with 1% glucose) to grow single and dual-species biofilms of the targeted microorganisms which, according to the Reviewer, is the “main drawback” of the manuscript.

We respectfully observe that a standard microbiological medium is used in most of the literature dealing with the study of single species and polymicrobial biofilms of pathogenic microorganisms and this is a “choice” rather than a “drawback”. For instance, in the list of references in our paper there are not less than 10 papers which demonstrate this fact (see for instance Reference 31: [Huffines, J. T., & Scoffield, J. A. (2020). Disruption of Streptococcus mutans and Candida albicans synergy by a commensal streptococcus. Scientific reports, 10(1), 1-10)]. 

Obviously, the medium can be enriched with components (biological fluids or tissue fragments, etc.) whose function is to simulate the biochemical environment of specific infected tissues or medical devices. 

Both approaches are legitimate, and the choice depends on the objectives pursued.

In our manuscript we consider for the first time the biological and metabolic changes in the interaction of C. glabrata-S. epidermidis during biofilm formation, and the scope of our work is to collect information on fundamental biological and metabolic effects of this interaction. Although we have selected the microorganisms considering their role in real-world infections, it is perfectly reasonable to consider first their interaction when they are cultivated in vitro in a standard medium which contains only the ingredients necessary to ensure wellness of both microorganisms in order to trace a groove for future research. 

Data collected under such conditions will not provide a definitive and universal model which can be applied without changes in all real-world situations but will give fundamental information which may be useful as a guide for further studies. In fact, in this field there is nothing like a universal model since any model is to some extend contingent and dependent on the data and conditions under which it was developed. The list of factors which may influence the model is very long and includes, among others, the strains of the microorganisms, the nature of the surface on which biofilms are deposited, the composition of the medium and the presence of exogenous substances which may or may not be present. This is because, in a specific work, factors which may influence the outcome of experiments must be accurately specified and controlled (as we did in our work). 

In other words, each model represents only one tooth of a gear and we do not pretend that our work represents the whole gear, but we do pretend that it represents a single tooth of the gear. 

The Reviewer suggestion of carrying out experiments using an environment simulating real-world infections is welcome since it is exactly what we are going to do in the continuing work, but it is, by definition, outside of the scope of the present work.

In any case we have accurately scrutinized the manuscript and made changes and added text and references to better specify its significance and further support our conclusions based on the presented data. 

- We have modified the title to “A Model for Microbial Interactions and Metabolomic Alterations in Candida glabrata-Staphylococcus epidermidis Dual-Species Biofilms” which makes evident that the outcome of our paper is a basic model for the interactions between C. glabrata/S. epidermidis biofilms.

 -By pursuing the same objective, we have substantially modified the “Introduction” and added text to better delineate the ambit and significance of our results.

- To better support our conclusions and to better connect them to previous studies, we have added further references and added text in the “Results” sections.

-We have also modified the “Conclusions” to make the significance of our work clearer.

---

## [Decision Letter · Decision Letter 2]

1 Dec 2022

A Model for Microbial Interactions and Metabolomic Alterations in Candida glabrata-Staphylococcus epidermidis Dual-Species Biofilms

PONE-D-22-20604R2

Dear Dr. Galdiero,

We’re pleased to inform you that your manuscript has been judged scientifically suitable for publication and will be formally accepted for publication once it meets all outstanding technical requirements.

Kind regards,

Marcos Pileggi, Ph.D

Academic Editor

PLOS ONE

Additional Editor Comments (optional):

Reviewers' comments:

Reviewer's Responses to Questions

**Comments to the Author**

1. If the authors have adequately addressed your comments raised in a previous round of review and you feel that this manuscript is now acceptable for publication, you may indicate that here to bypass the “Comments to the Author” section, enter your conflict of interest statement in the “Confidential to Editor” section, and submit your "Accept" recommendation.

Reviewer #2: All comments have been addressed

2. Is the manuscript technically sound, and do the data support the conclusions?

Reviewer #2: Yes

3. Has the statistical analysis been performed appropriately and rigorously? 

Reviewer #2: Yes

4. Have the authors made all data underlying the findings in their manuscript fully available?

Reviewer #2: Yes

5. Is the manuscript presented in an intelligible fashion and written in standard English?

Reviewer #2: Yes

6. Review Comments to the Author

Reviewer #2: (No Response)

7. PLOS authors have the option to publish the peer review history of their article (what does this mean?). If published, this will include your full peer review and any attached files.

Reviewer #2: No

---

## [Editor Report · Acceptance letter]

4 Dec 2022

PONE-D-22-20604R2 

A Model for Microbial Interactions and Metabolomic Alterations in *Candida glabrata-Staphylococcus epidermidis* Dual-Species Biofilms 

Dear Dr. Galdiero:

I'm pleased to inform you that your manuscript has been deemed suitable for publication in PLOS ONE. Congratulations! Your manuscript is now with our production department. 

Kind regards, 

on behalf of

Dr. Marcos Pileggi 

Academic Editor

PLOS ONE